# Spatial transcriptomics uncover sucrose post-phloem transport during maize kernel development

Yuxin Fu [1,2,3,6], Wenxin Xiao [1,6], Lang Tian[1], Liangxing Guo [2], Guangjin Ma[2,4], Chen Ji [2,4], Yongcai Huang [2,5], Haihai Wang [2], Xingguo Wu[1], Tao Yang [2,5], Jiechen Wang[2], Jirui Wang [3,5], Yongrui Wu [2] ✉ & Wenqin Wang [1] ✉

Maize kernels are complex biological systems composed of three genetic sources, namely maternal tissues, progeny embryos, and progeny endosperms. The lack of gene expression profiles with spatial information has limited the understanding of the specific functions of each cell population, and hindered the exploration of superior genes in kernels. In our study, we conduct microscopic sectioning and spatial transcriptomics analysis during the grain filling stage of maize kernels. This enables us to visualize the expression patterns of all genes through electronical RNA in situ hybridization, and identify 11 cell populations and 332 molecular marker genes. Furthermore, we systematically elucidate the spatial storage mechanisms of the three major substances in maize kernels: starch, protein, and oil. These findings provide valuable insights into the functional genes that control agronomic traits in maize kernels.

Maize is one of the most widely cultivated and high-yielding crops in the world, with an impressive average annual yield of up to 1 billion tons[1]. The kernel of maize is a complex system consisting of millions of cells, including diploid seed coats from maternal tissue, triploid endosperm, and diploid embryos from progeny genotypes. The embryo, which develops into the future plant, and the endosperm, which nourishes the seedling during germination, undergo three main phases of development[2,3], namely, the early stage of the first two weeks[4], the filling stage of the following two weeks[5], and the maturation phase of the last four weeks[6]. The accumulation of nutrients during the filling stage is crucial for kernel quality and yield, and it requires efficient coordination and harmonized allocation of fixed carbon, primarily in the form of sucrose, among intercompartment. In most crop plants, including maize, sucrose is transported over long distances through veins from mesophyll cells into sink tissues. Sucrose is either directly retrieved by sucrose transporters or hydrolyzed into

glucose and fructose, followed by subsequent uptake by hexose transporters and SWEETs through the narrow basal endosperm transfer layer (endosperm-BETL), which is the apoplasmic phloem unloading region adjacent to the maternal-filial interface[7].

However, the exact role of sucrose transport is not well defined due to the challenge of defining cell populations with a few cell layers and the cell similarity at the maternal-filial interface. Moreover, little is known about the number of cell populations that exist in maize, how these cell populations function, and how they coordinate with each other, which has become a bottleneck for exploring more genes to improve agronomic traits.

Significant advancements have been made in plant science to spatially quantify gene expression using next-generation sequencing approaches. RNA-seq analysis of embryo and endosperm tissues from mixed cell populations has revealed dynamic transcriptomes[8,9], but lost spatial information. Laser-capture microdissection (LCM)

[1]College of Life Science, Shanghai Normal University, 100 Guilin Road, Shanghai 200233, China. [2]National Key Laboratory of Plant Molecular Genetics, CAS Center for Excellence in Molecular Plant Sciences, Shanghai Institute of Plant Physiology & Ecology, Shanghai 200032, China. [3]Triticeae Research Institute, Sichuan Agricultural University, Chengdu 611130, China. [4]University of the Chinese Academy of Sciences, Beijing 100049, China. [5]State key Laboratory of Crop Gene Exploration and Utilization in Southwest China, Sichuan Agricultural University, Chengdu 611130, China. [6]These authors contributed equally: Yuxin Fu, Wenxin Xiao. ✉e-mail: yrwu@cemps.ac.cn; wang2021@shnu.edu.cn

facilitates the mRNA level profiling of dissected filial and maternal compartments, yielding a high-resolution expression map in maize. Nonetheless, this technique demands a prerequisite understanding and meticulous technical expertize[10,11]. Additionally, dissecting cells that resemble the surrounding tissue, such as regions of the basal endosperm transfer layer (endosperm-BETL), the endosperm adjacent to the scutellum (endosperm-EAS), and the embryo-surrounding region (endosperm-ESR), are relatively inaccessible and pose a risk of capturing unwanted cells due to contamination[12]. Although single-cell transcriptome atlases have been used to investigate cellular heterogeneity in plants, such as Arabidopsis vegetative shoots and rice roots[13,14], they do not directly resolve the spatial patterns of gene expression, heavily relying on known molecular markers to locate cell identity (Table S1).

Spatial transcriptomics represents a state-of-the-art profiling technique that enables the sequencing of total mRNA in fresh frozen tissues while simultaneously mapping these transcripts to their corresponding tissue positions using unique barcodes to identify spatial gene expression patterns. The first spatially resolved transcriptome profiling was demonstrated in the model plant species Arabidopsis thaliana, which revealed 141 differentially expressed genes across tissue domains. This methodology has subsequently been demonstrated to be more robust, powerful more sensitive than the experimental RNA in situ hybridization when applied to other plant species, including Populus leaf buds, Picea cones, and soybean nodules[15,16].

Given the complexity and incomplete understanding of maize kernels, we used spatial transcriptomics (ST) as an alternative method to distinguish morphologically similar but functionally distinct cell populations[17]. In this study, we present a comprehensive spatial organization of the cellular atlas that systematically describes the cellular heterogeneity of developing kernels (Fig. S1). We identified the obstinate cell populations of the basal endosperm transfer layer (endosperm-BETL), the embryo-surrounding region (endosperm-ESR), and the endosperm adjacent to the embryo (endosperm-EAS), and uncovered specific marker genes. Moreover, we have developed a web server that allows for the direct visualization of the genome-wide electronic RNA in situ hybridization map to facilitate gene annotation and superior gene mining.

## Results

### Kernel size and cell expansion during maize endosperm development

The maize inbred of W64A was selected for spatial transcriptomic study because it was extensively used in investigating gene functions associated with grain filling and endosperm texture during kernel development. To accurately assess the sample stage concerning kernel size, storage accumulation, and cell expansion, we conducted longitudinal free-hand sections and semi-thin sections of the W64A inbred at 10, 14, and 18 DAP. Our analysis revealed that kernel sections' size at 10–24 DAP was smaller than 4 × 8 mm (Figs. S2 and S3). As anticipated, endosperm cells stopped proliferating and started synthesizing storage metabolites after 10 DAP. The fate of cells in the kernel, including pericarp, aleurone (endosperm-AL), endosperm, basal endosperm transfer layer (endosperm-BETL), and other tissue organs, was determined at that time (Fig. S2)[18]. It is worth mentioning that the capture region on the spatial transcriptomic chip is a square with a side length of 6.5 mm that could accommodate a diagonally placed kernel, leading to the gene expression over the full grain sections to be easily observed (Fig. S3) and making maize kernels ideal materials for the study of spatial transcriptomics with a complete tissue section. Secondly, the chip incorporates 5000 spots containing arrayed oligonucleotide barcodes not only for mRNA capture but for spatial determination[19]. The tissue cells in maize kernels, in terms of maternal tissue, embryo, and endosperm, are of varying and uneven sizes. The average size of starchy endosperm cells at 10 DAP is 50 μm, and 60 μm at 14 DAP,

whereas the cells in the embryo range from 10 to 30 μm. The diameter of the spots on the 10x Genomics Visium chip is 55 μm, resulting in instances where a single spot can encompass multiple cells. The aim of this technique is to identify molecular markers and perform cell clustering to investigate their biological functions. We expect that the same functional region consists of multiple spots, which can be considered as biological replicates at the cellular level. This approach leads to more robust and reliable results, as it helps to minimize the impact of individual variations and increases the statistical power of the experiment (Fig. S1). Here, the stages of 12, 18 (two biological replicates), and 24 DAP representing the early, middle, and late filling stages were subjected to the spatial transcriptomic study. They were designated S12D, S18D_1, S18D_2, S24D in subsequent analyses.

### Spatial transcriptomes reveal kernel heterogeneity and functional compartments at filling stages

The kernels sampled from 12, 18, and 24 DAP were subjected to embedding, fixing, sectioning, and Hematoxylin–Eosin (HE) dying before spatial transcriptomic sequencing to overview the seed layout (Fig. S1). We found that the scutellum was colored gray and the embryo meristem (EM) was blue, indicating active cell division. The interface layers of endosperm-BETL and aleurone (endosperm-AL) that connect the filial tissue with the maternal tissue are shown in purple. The starchy endosperm cells filled with starch granules could not be stained, leading to dark gray, whereas the conducting zone (endosperm-CZ) next to endosperm-BETL was bright gray (Fig. 1a).

The cDNA library with the retention of spatial information after reverse transcription was sequenced using 10x Genomics Visium platform (Figs. S1 and S4). We utilized Unique Molecular Identifiers (UMI) to map the quantities of RNA molecules onto the sample sections. Our findings indicated that the missing tissue corresponded to the starchy region, characterized by a high concentration of starch. This region was represented by the dark blue color, indicating low expression levels of genes. However, it is important to note that there are still remaining sections of similar tissue type, which can potentially compensate for the missing information (Fig. S5). We detected an average of 10,901–12,257 unique transcripts and 3573–3912 expressed genes per spot, where the spots of embryo and aleurone contributed the most gene and transcript density (Fig. S5 and Table S2). We found that the biological replicates of 18 DAP showed the best Pearson correlation coefficient of 0.989, followed by other developmental stages (0.898–0.950) (Fig. S6), indicating that the bulk gene expression patterns at these stages were comparable.

We identified 25 spatial clusters based on the gene expression similarity using dimensional reduction (Fig. 1b). In order to validate the existence of new cell types, we have compiled a set of marker genes for all 25 cell clusters identified through our spatial transcriptomics analysis (Fig. S7). Additionally, we have included markers from the literature, which encompass in situ hybridization[11], RNA-seq from manual dissection[3], and RNA-seq from laser microdissection[11] (Table S3). It is noteworthy that nearly all cell populations were successfully validated using these alternative technologies. The clusters from 0 to 23 were shared across all stages except cluster 24 (Fig. S8 and Table S4). Integrated with the anatomical information from semi-thin sections and HE dyeing image (Fig. 1a and Fig. S2), we continued to merge these clusters into eleven functional cell populations (Fig. 1c). In this context, the term "cell populations" is also referred to as "compartments," representing groups of cells with similar gene expression patterns and physical proximity, indicating their similar functions, including 1) two maternal-derived regions: placento-chalazal (maternal-PC) and pericarp (maternal-PE); 2) two embryo regions: scutellum (embryo-SCU) and embryo meristem (embryo-EM); 3) seven endosperm compartments: basal endosperm transfer layer (endosperm-BETL), conducting zone (endosperm-CZ), endosperm adjacent to scutellum (endosperm-EAS), starchy endosperm (endosperm-SE), vitreous endosperm

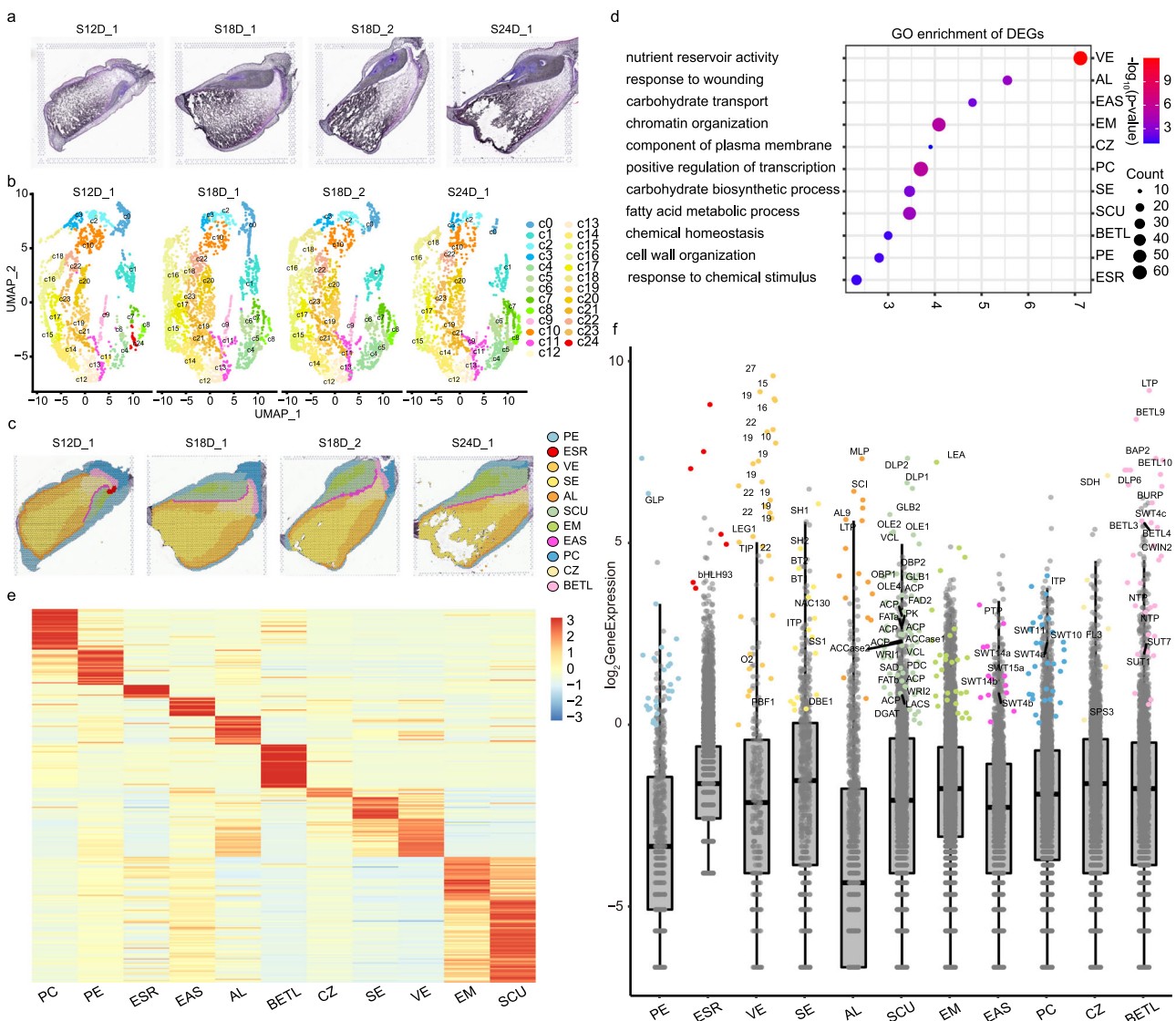

**Fig. 1 | Global spatiotemporal analysis of three filling stages. a** Histological structure of developing kernels at 12, 18 DAP, and 24 DAP. Side length = 6.5 mm. **b** Dimensional reduction and spot clustering. **c** Kernel compartment identification. **d** GO enrichment for differentially expressed genes. PC-GO:0045944, positive regulation of transcription by RNA polymerase II; PE-GO:0071554, cell wall organization or biogenesis; SCU- GO:0006631, fatty acid metabolic process; EM-GO:0006325, chromatin organization; BETL-GO:0048878, chemical homeostasis; AL-GO:0009611, response to wounding; EAS-GO:0008643, carbohydrate transport; CZ-GO:0046658, anchored component of membrane; SE-GO:0016051, carbohydrate biosynthetic process; VE-GO:0045735, nutrient reservoir activity; ESR-GO:0070887, cellular response to chemical stimulus. The enriched GO categories were determined using the one-sided version of Fisher's exact test, followed by the Benjamin-Hochberg correction to obtain adjusted *p* values for multiple testing. **e** Heat map of scaled marker gene expression (UMI index). The color scale from blue to red represents the UMI index. **f** The marker gene expression level compared

with non-marker genes. The marker genes are labeled with different colors according to compartments. Due to space limitations, only the genes involved in plant defense, seed development, sugar transport, starch, protein, and lipid accumulation are labeled. The non-marker genes are colored in gray dots. The box in the boxplot represents the middle 50% of the data, which is equivalent to the interquartile range (IQR, the difference between the 25th percentile and the 75th percentile) The median value, which is the second quartile, is positioned in the middle, dividing the data into two halves. Each whisker extends to the farthest data point within 1.5 times the IQR in each direction. The lower end of the dark line represents the minimum gene expression, while the upper end represents the maximum gene expression. Abbreviation is provided in Supplementary Data 4. S12D_1 indicates the sample taken at 12 DAP; S18D_1 indicates the sample taken at 18 DAP; S18D_2 indicates the biological sample taken at 18 DAP; S24D_1 indicates the sample taken at 24 DAP. Source data are provided as a Source Data file.

(endosperm-VE), aleurone (endosperm-AL), and embryo-surrounding region (endosperm-ESR). The three interfaces including the endosperm-BETL between maternal and filial tissue, the endosperm-EAS between embryo and endosperm, and the endosperm-AL between maternal and endosperm were accurately located. Moreover, endosperm-ESR corresponding to cluster 24 was only represented at 12 DAP and disappeared at 18 and 24 DAP (Fig. 1c)[20]. Spatial transcriptomics enables the simultaneous identification of RNA in situ expression data for approximately 40,000 genes throughout the

entire maize genome. This high throughput capability enables heightened sensitivity to uncover novel cell types. We found that the pericarp encompasses three distinct cell types, the embryo can be categorized into five new groups, the starchy endosperm can be classified into five groups, and the vitreous endosperm can be further divided into five groups (Fig. 1b and Fig. S8).

There were 26,161 genes expressed in at least one of the eleven compartments after pooling all four sections (Supplementary Data 1 and Fig. S9). To construct the gene co-expression network, we

extracted 2986 genes from the initial pool of 26,161 genes by excluding those with low expression levels (UMI < 1) or low variability (coefficient of variation <0.8) across different cell types that usually represent noise. Subsequently, we performed a weighted gene correlation network analysis (WGCNA) and identified eleven co-expression modules (Supplementary Data 2 and Fig. S10a). We found that the individual module was closely related to eleven corresponding anatomical regions via the correlation analysis (Fig. S10b). The GO functional annotations of modules were enriched in certain biological process, including carbohydrate transport, starch biosynthetic process, fatty acid biosynthetic process, nutrient reservoir activity, et al. (Table S5), consistent with the filling stage of developing kernels.

Evidence showed that the compartments with the highest overlap in mRNA populations are expected to be more closely associated and are likely to share functions[21]. The hierarchical clustering from the Pearson correlation (Fig. S11) showed the compartments from nearby anatomical tissues were closely correlated, where three large functional regions were displayed: (1) storage accumulation (endosperm-AL, endosperm-SE, endosperm-VE, endosperm-CZ, endosperm-EAS, and maternal-PE); (2) offspring development (embryo-SCU and embryo-EM); (3) maternal-filial interface (endosperm-ESR, endosperm-BETL, and maternal-PC). We discovered that maternal-PE was closer to the endosperm compartments, while maternal-PC showed a stronger correlation with endosperm-BETL. This suggests that the gene expression programs in maternal and filial tissues are cooperatively functioning, despite their divergent genetic origins.

## Mining key genes essential for sucrose transport and storage accumulation

We defined 2,992 differentially expressed genes, presenting higher expression in a specific compartment, but lower expression in other compartments (Supplementary Data 3). Gene ontology analyses suggest that the endosperm-SE, endosperm-VE, and embryo-SCU display carbohydrate biosynthetic process, nutrient reservoir activity, and fatty acid metabolic process, respectively (Fig. 1d, f). A further selection of 332 genes were defined as the marker genes with the features of significant gene-gene connectivity in WGCNA analysis, which belonged to the members of differentially expressed genes and had a minimal expression level of 1 (UMI index) (Fig. 1e and Supplementary Data 4). They were visualized using spatial transcriptomic data overlaid on tissue sections, producing a powerful electronic RNA in situ hybridization (Fig. 2 and Fig. S12). The known markers from previous published data[10,22], for example, *AL9* in the aleurone (Zm00001d012572), *CWIN2* in the endosperm-BETL (Zm00001d003776), *SWEET14a* in the endosperm-EAS (Zm00001d007365), *ACCase* in the embryo-SCU (Zm00001d004125), showed highly specific mRNA localization patterns within the corresponding cell types. These data indicated that our spatial transcriptomic data accurately reflected the localization of endogenous mRNAs in the kernel (Fig. S12). In addition, the defined markers clearly showed cellular heterogeneity, such as Zm00001d036784 encoding a potassium transporter in the endosperm-BETL, Zm00001d031727 encoding a sorbitol dehydrogenase in the endosperm-CZ (ranked first in the endosperm-CZ and 25th in all expressed genes, Supplementary Data 1), Zm00001d018445 encoding a phosphate transporter in the endosperm-EAS, Zm00001d011340 encoding the acanthoscurrin in the endosperm-ESR, and Zm00001d002952 encoding the aquaporin in the embryo-SCU et al (Fig. S13). These identified marker genes would help reveal the novel functions of the specific cell populations. To validate the markers we have identified, we conducted a comparison with a previous study that utilized laser-capture microdissection (LCM) to profile mRNA populations within the primary cell types[10]. By comparing their markers with ours, we observed a consistent overlap, indicating the accuracy and reliability of our methodology (Table S6). Furthermore, we randomly

selected twelve defined marker genes to validate the mRNA distribution using RNA in situ hybridization (Fig. 2 and Table S7) and they showed specific expression signals in their corresponding compartments, consistent with the spatial transcriptomic data (Fig. 2). Contrastingly, while some of the RNA in situ hybridizations exhibited unclear signals due to the noisy background probably resulting from over-staining or non-specific hybridization in the experiments (Fig. 2g, o, q s), the electronic RNA in situ hybridization maps created from the spatial transcriptomics were more specific and clearer (Fig. 2).

As we look closer to the list of marker genes after computational and manual annotations (Supplementary Data 4 and Fig. 1f), we found that many members in some gene families tended to express in specific compartments, such as *MADs* (twelve members) in the maternal-PC, *oleosin* (eight members) in the embryo-SCU, *zeins* (twenty-one members) in the endosperm-VE and *globulins* (three members) in the embryo (Supplementary Data 4 and Fig. 1f), indicating they have specific functions in kernel development and grain filling. A few transcription factors from bHLH, MYB, and GRAS families were identified as marker genes, indicating they play pivotal roles in kernel development. However, their biological functions in regulating kernel development require further experimental validations (Supplementary Data 4).

## Spatially differential distribution of storage metabolite synthesis-related transcripts in maize seeds

Maize endosperm consists of vitreous endosperm and starchy endosperm (Fig. 3a). The kernel texture is influenced by the ratio of vitreous endosperm in the peripheral region of the kernel to the starchy endosperm in the center region. Vitreous endosperm makes the kernel harder and protects the grains from mechanical damages during harvesting, transportation, and storage, whereas starchy endosperm is breakable and fragile, and is susceptible to pests and diseases. Many factors affect vitreous endosperm formation, of which the synthesis of starch and zein proteins plays essential roles in this process. During grain filling, the starch granules are surrounded by protein bodies in the cytoplasm. However, starch granules in the periphery of the endosperm are fewer and smaller than those in the center, while protein bodies are the inverse for number and size in these two regions. At seed maturity, condensation of these components leads to the formation of vitreous endosperm in the outer area, whereas the inner area forms soft, starchy endosperm (Fig. 3b). However, the mechanisms for differential synthesis of zein proteins and starch in the two areas remain unclear. We found that the a number of genes encoding enzymes for starch biosynthesis[10,23] including the sucrose synthase (SH1), glucose-1-phosphate adenylyltransferase large and small subunits (SH2 and BT2), adenine nucleotide transporter (BT1), starch synthases (SS1 and SS2), starch debranching enzymes (DBE1) and sugary 2 (SU2) as marker genes in the endosperm-SE region (Figs. 1f and 3c and Supplementary Data 4), consistent with the endosperm-SE being located in the central part of the kernel (Fig. 3a). We also found that 21 *zein* members were ranked as the top expressed genes in the endosperm-VE, including the *15-kD β-zein*, the *16-kD γ-zein*, the *27-kD γ-zein*, and the *19-kD* and *22-kD α-zeins* (Figs. 1f and 3d and Supplementary Data 4). *18-kD δ-zein* is a mutant gene in the W64A inbred[24], but its transcripts were also detected in the endosperm-AL and endosperm-VE. The *50-kD γ-zein* was highly expressed both in endosperm-SE and endosperm-VE (Supplementary Data 4). The spatially differential expression of *zein* genes in the endosperm supported the fact that the vitreous endosperm is formed in the peripheral region[25].

Most of the oil and lipid in maize seeds accumulate in the embryo (Fig. 3e). We found that the genes encoding a few key enzymes in fatty acid biosynthesis consisting of acetyl-CoA carboxylase for substrate activation, acyl carrier protein (ACP) for elongation, fatty acid desaturase for desaturation were highly expressed in the embryo-SCU and

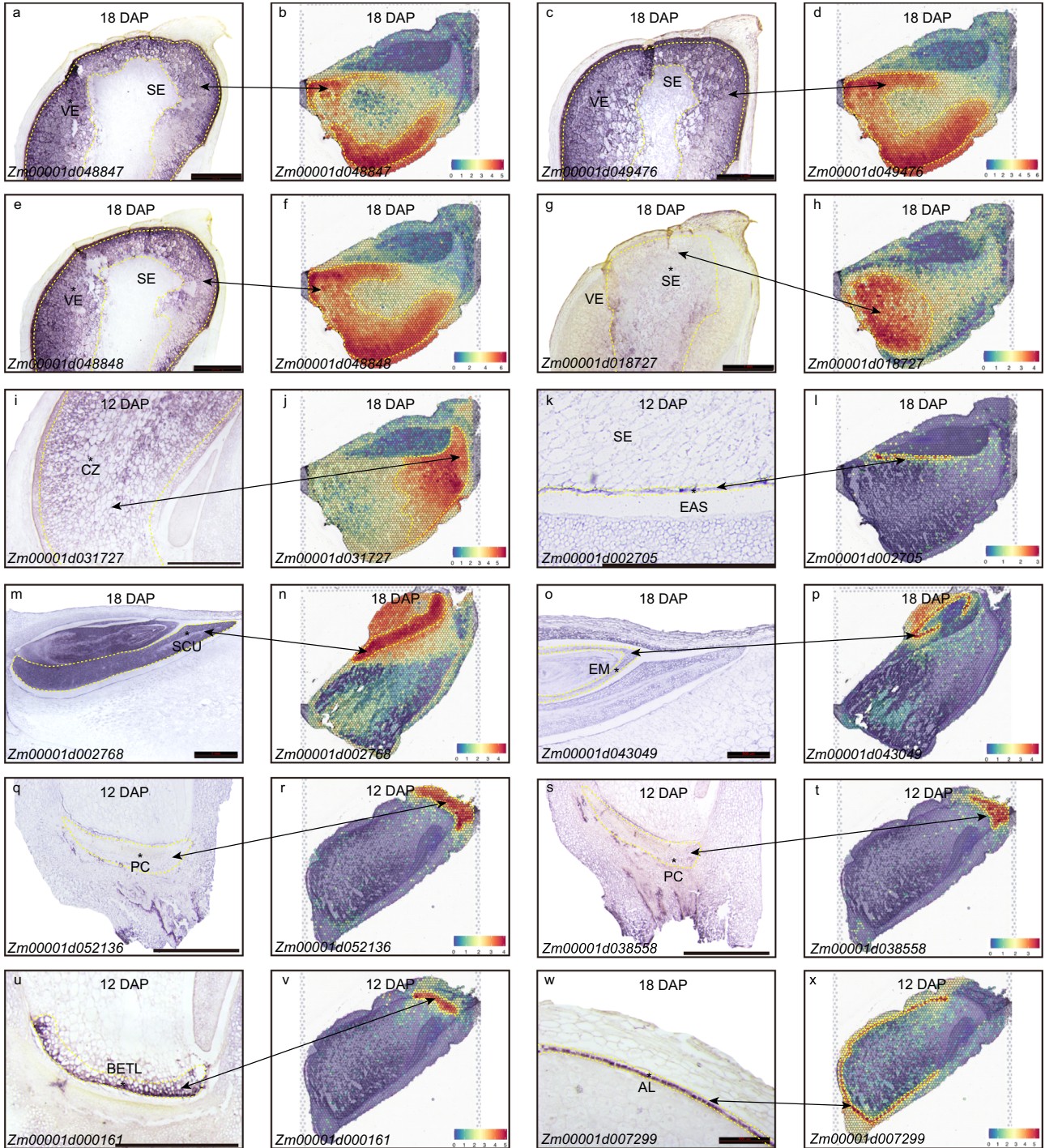

**Fig. 2 | Marker gene validation using experimental and electronical RNA in situ hybridization.** Line 1 and 3 in vertical (**a**, **c**, **e**, **g**, **i**, **k**, **m**, **o**, **q**, **s**, **u**, **w**) show the experimental in situ hybridization. Longitudinal sections of W64A kernels at 12, 18 DAP were hybridized with antisense RNA probes. The hybridization signals of the marker genes appeared in blue violet in kernels. Three genes of in **g**, **q**, **s** present weak signals due to the low expression or noise background. Scale bars is 1 mm.

The experimental in situ hybridization for each gene was independently repeated three times. Each repetition included a minimum of 15 cross-sections of seeds. Line 2 and 4 in vertical (**b**, **d**, **f**, **h**, **j**, **l**, **n**, **p**, **r**, **t**, **v**, **x**) represent electronical gene expression using spatial transcriptomic data corresponding to the experimental markers. Source data are provided as a Source Data file.

coincided with the embryo as an oil storage organ (Fig. 1f). We found that some genes involved in lipid biosynthesis were highly expressed in the embryo-SCU, such as Zm00001d045988, Zm00001d002768, and Zm00001d033612 encoding oleosin, and Zm00001d043464, Zm00001d051459, and Zm00001d011755 encoding the oil body-associated protein (Figs. 1f and 3f). These results are consistent with spatially specific accumulation of oil and lipid in maize seeds[2].

## Functional validation of sucrose transporter genes

In the process of grain filling, sucrose is generally believed to be transported from the source to the sink tissue. After unloading in the phloem region, sucrose is either cleaved into fructose and glucose by cell wall invertase that is uptake by monosaccharide transporters, or sucrose is directly retrieved by sucrose transporters (SUTs) in the small gateway of kernels for nutrient delivery. Previous studies have shown

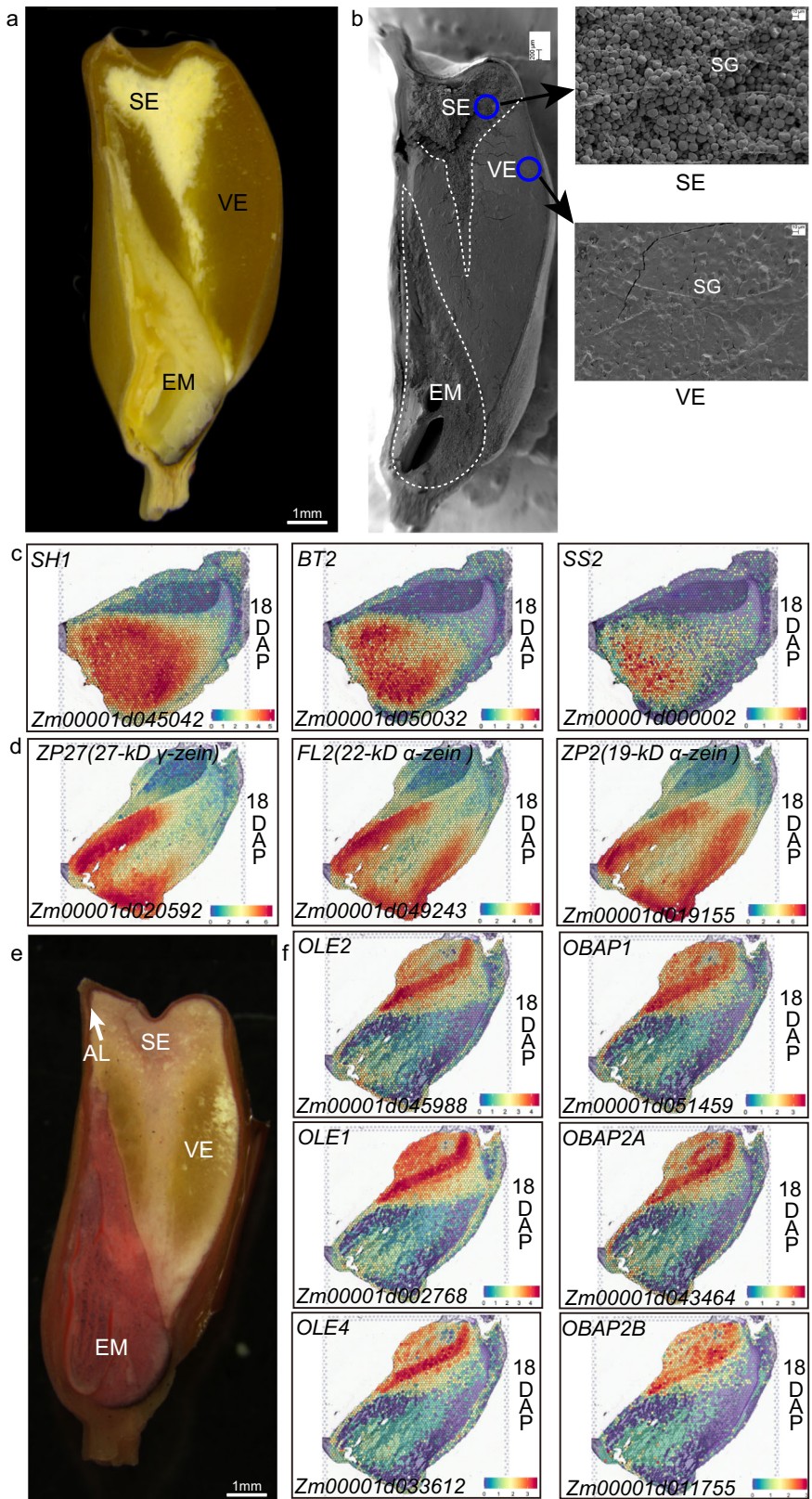

**Fig. 3 | Spatially specific accumulation of starch, proteins, and oil in maize seeds. a** A longitudinal section of the mature W64A kernel. The section was observed under the natural light. Bar = 1 mm. **b** Scanning electron microscopy of the longitudinal section of the mature W64A kernel. The dotted lines mark the areas of the vitreous endosperm, starchy endosperm, and embryo. The blue circle shows the location of the enlarged image on the right panel. VE vitreous endosperm, SE starchy endosperm, EM embryo, SG starch granule. Bars = 200 μm. The experiment was independently repeated three times. **c** Spatial expression patterns of starch synthesis-related genes in maize kernel. **d** Spatial expression patterns of *zein* genes in maize kernel. **e** Sudan IV dyeing of the longitudinal section of the mature W64A kernel. The red staining mainly occurred in SCU. The section was observed under natural light. Bar = 1 mm. **f** Spatial expression patterns of oil synthesis-related genes in maize kernel. Source data are provided as a Source Data file.

that SUTs play critical functions in sucrose phloem loading in maize leaves[26]. However, the exact role of SUTs in the apoplasmic step of post-phloem transport remains unknown due to the lack of knowledge of their spatial expression and localization[7].

We found there were seven genes encoding sucrose transporter in the maize genome, including *ZmSUT1* (Zm00001d027854), *ZmSUT2* (Zm00001d041192), *ZmSUT3* (Zm00001d033011), *ZmSUT4* (Zm00001d018527), *ZmSUT5* (Zm00001d016938), and *ZmSUT6* (Zm00001d050873) and *ZmSUT7* (Zm00001d048311). ZmSUT1 and ZmSUT7 were highly conserved with the identity of 93% in protein sequences and 91% in cDNA coding sequences, indicating that they were derived from a recent duplication (Figs. S14 and S15). qRT-PCR revealed that *ZmSUT1, ZmSUT2, ZmSUT4, ZmSUT5,* and *ZmSUT7* were expressed at different levels in maize seeds (Fig. S16 and Table S8). Our spatial transcriptome data showed that only *ZmSUT1* and *ZmSUT7* were specifically expressed in the BETL, indicating the two duplicated copies are conserved in spatial regulation (Fig. 4a), while *ZmSUT2, ZmSUT4,* and *ZmSUT5* had no specific spatial pattern in maize seeds.

To validate the function of *ZmSUT1* and *ZmSUT7* in grain filling, we specifically suppressed *ZmSUT1* and *ZmSUT7* expression via RNA interference (RNAi) driven a BETL-specific gene (*Betl9*) promoter and recovered two independent RNAi transgenic lines. We found that the cell wall ingrowth (CWI) in the BETL region of *zmsut1/7RNAi* seeds was apparently inhibited, indicating *ZmSUT1* and *ZmSUT7* are essential for CWI formation (Fig. 4b). The kernel size and weight of the self-pollinated *zmsut1/7RNAi* transgenic lines were greatly reduced compared with the control (Fig. 4d, e). To exclude the possible influence of *zmsut1/7RNAi* on plant growth, which in turn affects seed development and grain filling, we used *zmsut1/7RNAi* pollen to pollinate the wild-type ears and found the seed size and weight of the resulting progeny seeds were also significantly reduced (Fig. 4c–e), confirming that *ZmSUT1* and *ZmSUT7* are crucial for grain filling in maize. Therefore, mutations occurring in *ZmSUT1/7* genes in the BETL region not only affect the phenotype of that particular area, but also lead to issues in the overall phenotype of the grain due to disruptions in nutrient transport.

## Discussion

Maize kernel development originates from double fertilization and mainly includes cell division and differentiation (1-10 DAP), storage accumulation (10-35 DAP), and maturation stages (35-56 DAP). During the early stage, the fertilized ovule undergoes rapid cell division and differentiation, but significant storage compound accumulation hasn't occurred yet. After 10 DAP, cell fate is largely determined by the ceasing of cell differentiation, and the kernels start the accumulation of storage compounds such as starch granules, protein bodies, and oils. We look forward to future projects aimed at understanding cell trajectories, where it will be crucial to examine samples taken earlier than 10 DAP, when cell fate remains undetermined.

After 10 DAP, the endosperm becomes the main storage organ for starch and protein accumulation, compared to the embryo which functions as the oil warehouse. Understanding the functions of cell populations that orchestrate storage substance accumulation requires knowing how the genes interact with a spatial context. Here, we present an innovative approach that takes the power of spatial transcriptomics technology to comprehensively characterize the genome-wide gene expression with spatial information during maize filling stage. We obtained gene transcriptional snapshots from three filling stages (early, middle, and late), enabling us to understand the kernel development. We identified a comprehensive set of gene markers that could summarize the spatial complexity of the developing kernels, leading to 11 functional compartments. Each distinct anatomical organ performs the specific function, but interweaves to accomplish the mission of reproduction. The acquisition of endosperm storage capacity is enabled through the activity of specialized cell types of

maternal-PC and endosperm-BETL that mediate uptake of nutrients from the maternal structures, which assure the downstream of starch and protein accumulation in endosperm, as well as lipid storage in embryo. A notable finding is that endosperm-ESR cells exhibit distinct spatiotemporal patterns in the early stage of kernel-filling development. Our finding suggested that endosperm-ESR cells are required for defense of the embryo and signaling between the embryo and the endosperm, consistent with the previous study[27]. We have the priority to distinguish the cell population of endosperm-EAS from the surrounding tissues in spite of the highly similar cell morphology. Surprisingly, a couple of SWEET family genes were highly expressed in the interface of endosperm and embryo (endosperm-EAS) rather than in endosperm-ESR, indicating the sucrose might be transport through endosperm-EAS aiming to nourish the embryo.

In conclusion, we explored global spatial transcriptional patterns in maize kernel using the technology of spatial transcriptomics, deconvolved their cellular heterogeneity, and determined critical genes that are responsible for cell-type differences. Our electronic RNA in situ hybridization map identified that *ZmSUT1* and *ZmSUT7* were specifically expressed in the endosperm-BETL region, while other *ZmSUT* genes were not. We genetically confirmed their essential functions in grain filling. All in all, we have created a publicly available web resource that can be used to visualize electronic RNA in situ hybridization over the whole kernel sections (http://119.78.67.206:3838).

## Methods

### Plant growth

The maize inbred line W64A was cultivated in greenhouse with a temperature of 28 ± 4 °C during pollination and kernel-filling stage. The developing kernels of 10, 12, 14, 18 DAP, and 24 DAP (days after pollination) were harvested, respectively.

### Tissue section analysis

The samples were fixed in FAA buffer (formaldehyde:acetic acid:ethanol:water = 10:5:50:35 [v/v/v/v]) and kept in a vacuum vessel for 30 min at 4 °C. The samples were embedded in epoxide resin for semi-thin sectioning after ethanol dehydration. The sections were stained with 0.1% toluidine blue solution and then photographed under Brightfield using a ZEISS Axio Zoom V16 microscope or a Leica DM2500 for a magnifying view.

### Hematoxylin & Eosin staining and imaging

The kernels sampled from 12, 18 (two biological replicates), and 24 DAP were embedded in OCT medium (SAKURA Tissue-Tek O.C.T Compound, LOT: 4583), snap frozen, and kept at −80 °C. The samples were longitudinally cryosectioned at 10-μm thickness and systematically mounted onto the 10× Visium chip, where contained four capture areas with a side length of 6.5 mm. Each square area includes 5000 barcoded spots with a diameter of 55 μm. Each spot contains millions of capture oligonucleotides that are unique to that spot. The samples were heated to 37 °C for 1 min and fixed for 30 min with methanol at −20 °C. They were dehydrated with isopropanol for 1 min followed by staining with hematoxylin and eosin. The chip was cleaned using ultrapure water and then left to dry at room temperature. Brightfield imaging was performed to check if the section covered the targeted region and to record the morphological information under the 3D HISTECH Pannoramic MIDI FL at ×40 resolution.

### Library construction and sequencing

The time of permeabilization for fresh frozen tissues was optimized to 20 min, allowing to release of the maximal mRNA that then bound to spatially barcoded spots. cDNA was then synthesized from the captured mRNA and prepared into a sequencing library following the manufacture manual (10×Genomics Visium platform, Demonstrated protocol CG000160).

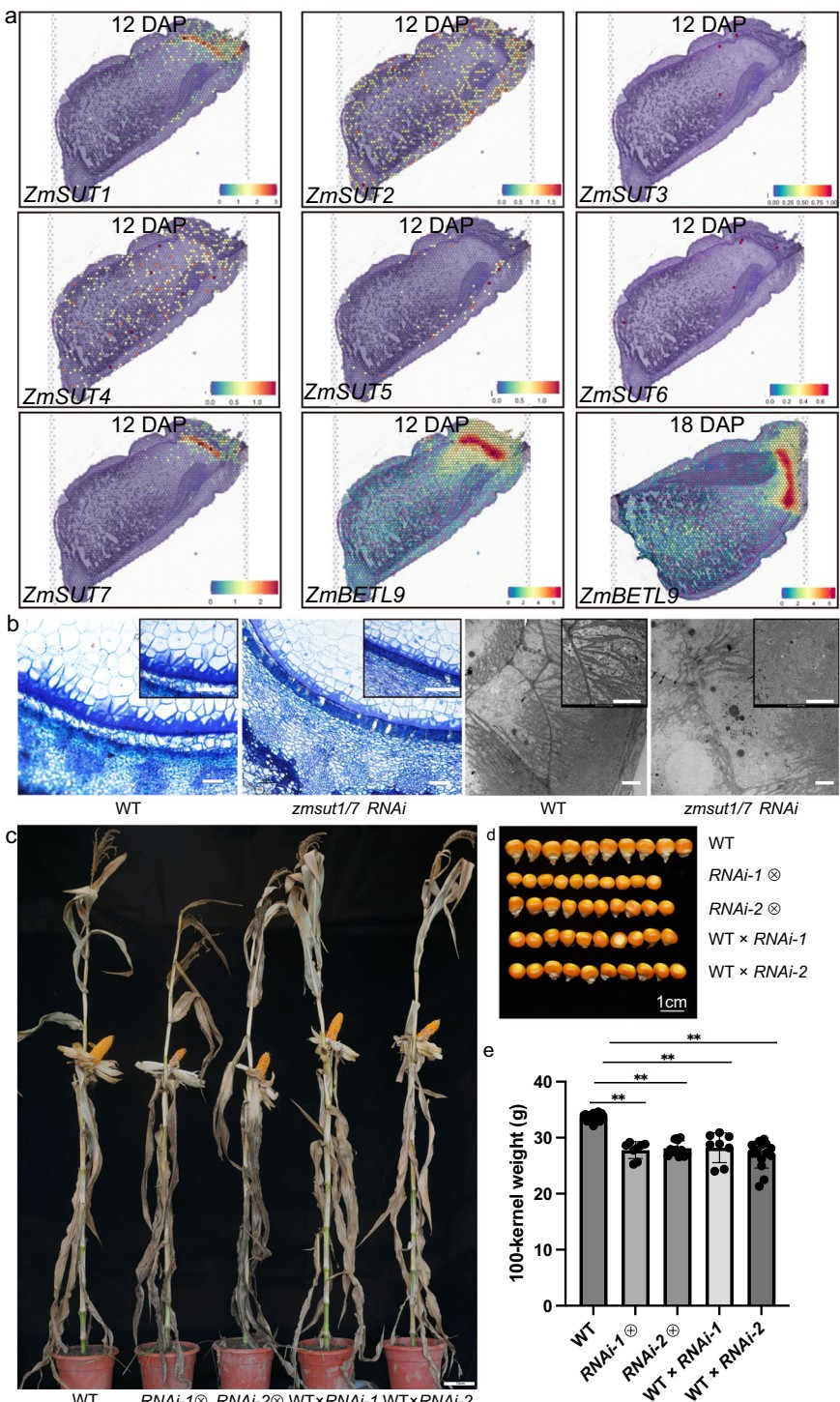

**Fig. 4 | Functional study of *ZmSUT* genes in maize seed. a** Spatial expression patterns of *ZmSUT* (14 DAP) and *Betl9* (14 and 18 DAP) genes. **b** Semi-thin sections (panels 1 and 2) and transmission electron microscopy (TEM, panels 3 and 4) of the BETL region in the wild-type KN5585 (panels 1 and 3) and *zmsutRNAi-1/7* (panels 2 and 4) seeds at 14 DAP. The experiment was independently repeated three times. In panels 1 and 2, bars = 100 μm; in panels 3 and 4, bars = 5 μm. **c** Plant phenotypes of the wild-type KN5585, selfing *zmsut1/7RNAi-1*, selfing *zmsut1/7RNAi-2*, WT × *RNAi-1* and WT × *RNAi-2*. Bar = 10 cm. **d** Kernel phenotypes of the wild-type KN5585, selfing *RNAi-1*, selfing *RNAi-2*, WT × *RNAi-1* and WT × *RNAi-2*. Bar = 1 cm. **e** 100-kernel weight of the wild-type KN5585, selfing RNAi-1, selfing RNAi-2, WT × RNAi-1 and WT × RNAi-2. A one-way ANOVA revealed that there was a statistically significant difference among the different genetic background between at least two groups (F(between groups df = 4, within groups df = 60) = 44.89, *P* = 2.15e-17). The further Tukey's HSD Test for multiple comparisons found that the mean value of 100-kernel weight was significantly different between the mutants and the wild type (**\**P* < 0.001). The details are: selfing RNAi-1 (means ± SD = 27.77 ± 1.50, *P* = 3.58289E-15, *n* = 8), selfing RNAi-2 (means ± SD = 28.12 ± 1.34, *P* = 3.90318E-17, *n* = 12), WT×RNAi-1 (means ± SD = 28.21 ± 2.65, *P* = 6.41384E-10, *n* = 8), WT×RNAi-2 (means ± SD = 27.00 ± 2.43, *P* = 3.02368E-14, *n* = 15), and WT (means ± SD = 33.62 ± 0.62, *n* = 22). Data are presented as means ± SD. *n* indicates the number of biologically independent samples. There was no statistically significant difference within mutant groups (*P* = 0.45–0.99). Source data are provided as a Source Data file.

## Sequence alignment, normalization, and quantification

After quality control, the raw reads were demultiplexed, assigned to their derived spots, and generated a feature-spot matrix according to the barcode oligonucleotides and brightfield microscope images using Space Ranger (https://www.10xgenomics.com). The B73_RefGen_v4 reference genome was downloaded from ENSEMBL (http://ftp.ensemblgenomes.org/) and the gene expression was counted based on the B73_RefGen_v4 annotation[28].

In our study, we utilized the Seurat R package developed and maintained by the Satija lab (https://satijalab.org/seurat/) to define cell types. Seurat has gained widespread popularity in the field of spatial transcriptomics due to its robustness, scalability, and user-friendly interface. It offers various modules that facilitate the analysis workflow, enabling researchers to perform tasks such as quality control, normalization, dimensionality reduction, clustering, and visualization. Here, we leveraged specific Seurat modules. First, the gene expression data from spatial spots were normalized and quantified using the Seurat R package[29]. To account for technical artifacts while preserving biological variance, we employed the SCTransform algorithm[30]. Principal component analysis (PCA) was performed to reduce the dimensionality on the log-transformed gene-barcode matrices of top variable genes. To cluster the tissue cells, we utilized a graph-based clustering approach combined with prior knowledge. The Seurat R package functions of RunPCA, FindNeighbors, and FindClusters were employed for dimensional reduction and clustering. To visualize the spatial gene expression within the tissue sections and map it to the morphological context, we used the SpatialDimPlot function from Seurat. This allowed us to gain insights into the spatial distribution patterns of gene expression. In the sequencing library preparation, Unique Molecular Identifiers (UMIs) were attached to RNA molecules. This process creates a distinct identity for each input molecule and corrects for PCR artifacts[31]. Molecules sharing the same UMI are assumed to originate from the same input molecule. In our study, the UMI index was defined as the number of expressed transcript molecules for a specific gene divided by the total transcripts in a spot, distinguished by UMI, and multiplied by a scale factor of 10,000. This calculation helped quantify the gene expression levels in each spatial spot accurately.

## Identification of differentially expressed genes and construction of co-expression network

The FindMarkers module in Seurat R package was used to find differentially expressed genes within anatomical regions (settings: min.pct = 0.25, thresh.use = 0.25, $P$ value < 0.05, and |$\log_2$foldchange|> 0.58 was set as the threshold for significantly differentially expressed genes)[29]. The dataset comprising 26,161 genes was filtered with low expression levels and lack of variability that was typically considered as noise when constructing the gene co-expression network. The threshold for the minimum gene expression level was set at 0.3, while the coefficient of variation was set at 0.8. The gene co-expression network was established using the R package of WGCNA[32]. We selected a soft-thresholding power of eight and a minimum gene number of 80 to identify highly correlated genes. The gene connectivity GS was determined for each module. The further functional enrichment analysis was conducted to obtain insights into the module functions. The Gene Ontology analysis for differentially expressed genes and WGNCA module genes were enriched using enrichGO with the cutoff of $P$ < 0.05 and clusterProfiler in R package, as well as the maize database captured by annotationHub[33]. We performed multiple test corrections to account for the potential issue of false positives. Specifically, we used established methods such as the Bonferroni correction or the Benjamini-Hochberg procedure to adjust the p-values obtained from the enrichment analysis.

## Marker genes identification and RNA in situ hybridization

The marker genes were further defined with significant gene-gene connectivity in WGNCA analysis (i.e., gene significance greater than 0.4 and weighted correlation index more than 0.4), being differentially expressed genes with a minimal expression level of 1 (UMI index). The electronics in situ hybridization generated by spatial transcriptomics was visualized using Seurat R module of SpatialFeaturePlot[29]. The experimental validation of marker genes using RNA in situ hybridization was done following the previous work[34], i.e., the kernel was fixed in 4% paraformaldehyde solution (Sigma) with 0.1% TritonX-100 (Sigma) and 0.1% Tween-20 in PBS (Takara, cat #900) for 16 hours. After dehydration using gradient ethanol and vitrification using xylene, the samples were embedded in paraffin. The kernel sections were cut into 10 μm longitudinally using Leica manual microtome (Leica FM2235). The fragment of the gene coding sequence was cloned and inserted into the pEasy-blunt-zero vector. The antisense and sense RNA probes were transcribed in vitro by T7 and SP6 RNA polymerase according to the instructions for the DIG RNA labeling kit (Roche, catalog number 11175025910). The primers were synthesized at Biosune Biotech (Shanghai, China). The primer sequences used in this study are shown in Table S7.

## Real-time quantitative reverse transcription PCR (qRT-PCR)

After the total RNAs were extracted from various tissues including root, stem, leaf, ear, seed, embryo, and endosperm, they were reverse transcribed into cDNA. The qRT-PCR was performed to quantify SUT gene expression using the kits of TBGreenTM Premix Ex TaqTM and CFX ConnectTM Real-Time System (Takara). The actin gene was the internal control. The primer sequences used in this study are shown in Table S8.

## Creation of RNAi transgenic plants of *ZmSUT1/7*

According to the *ZmSUT1* (Zm00001d027854) and *ZmSUT7* (Zm00001d048311) cDNA sequence alignment, we used the conserved sequences to create an RNAi cassette driven by the endosperm-BETL-specific gene (*Belt9*, Zm00001d04182) promoter to suppress both *ZmSUT1* and *ZmSUT7* expression. This cassette was transferred into the pTF102 construct, which was then used to transform the KN5585 inbred. Two independent *zmsut1/7RNAi* transgenic events were recovered. The transgenic plants were self-pollinated for two generations to obtain homozygous ears. The heterozygous ears were obtained by crossing the wild type and homozygous *zmsut1/7RNAi* plants. The constructs were transformed into KN5585 by Wimi Biotechnology (Jiangsu, China).

## Transmission electron microscope analysis

Fresh wild-type and transgenic ears were collected 18 days after pollination. A small piece of kernel was cut from the endosperm-BETL area and fixed in a 0.2 M phosphate buffer (pH 7.2) containing 2.5% glutaraldehyde. The sample was then fixed in 2.5% osmic acid, dehydrated, and embedded in epoxy resin. Ultrathin sections were obtained using Leica microtome, stained with uranyl acetate and lead citrate, and observed under a transmission electron microscope (Hitachi H-7650). Mature corn kernels were longitudinally cut, sputtering with platinum for 100 seconds (QUORUM Q150T), observed under field emission scanning electron microscopy (Zeiss Merlin Compact).

## Statistics and reproducibility

No statistical method was used to predetermine the sample size. No data were excluded from the analyses. The experiments were not randomized. The Investigators were not blinded to allocation during experiments and outcome assessment.

## Reporting summary

Further information on research design is available in the Nature Portfolio Reporting Summary linked to this article.

## Data availability

The raw sequencing data of spatial Transcriptomics generated in this study have been deposited in the NCBI Sequence Read Archive under the BioProject of PRJNA1031340 with the accession code SRX22238644-SRX22238647. The electronic RNA in situ hybridization images are available to the community using R package of Seurat for visualizing gene expression over the whole tissue sections of 12, 18, and 24-DAP maize developing kernels. (http://119.78.67.206:3838; see Fig. S12 for examples). The source data underlying Figs. 1–4 are provided in the Supplementary Data files and Source Data file. Source data are provided with this paper.

## Code availability

Details of the spatial transcriptomic analysis pipeline and R coding scripts can be found on our GitHub repository (https://github.com/wwq413/SpatialTranscriptomics)[35].

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

## Acknowledgements

We thank Hongyan Liu, Keke Yin, Man Jiang, Xue Song for bioinformatic assistance, sample frozen-section, and library construction from OE Biotech Co., Ltd (Shanghai, China). We thank Dr. Tao Huang for computer infrastructure support. We thank Zhiping Zhang (CAS Center for Excellence in Molecular Plant Sciences, SIPPE) for their technical support for the electron microscope. We thank Dr. Hongjun Liu from Shandong Agricultural University for maize planting. This work was

supported by the National Science and Technology Major Project of the Ministry of Science and Technology of China (2022YFF1003302 to W.W. and Y.W.) and the National Science Foundation (32072008, 32372068 to W.W. and 31830063, 31925030 to Y.W.).

## Author contributions

W.W., Y.W., and J.R.W. designed the study, analyzed the data, interpreted the results, and wrote the manuscript. Y.F. performed tissue sectioning, S.T. sequencing experiments, RNA in situ hybridization, and analyzed the data. W.X. and L.T. conducted a bioinformatic analysis and presented the data. G.M. and X.W. provided bioinformatic support and maintained the database. L.G., C.J., Y.H., T.Y., H.W., and J.C.W. generated transgenic lines, investigated phenotypes, and performed molecular experiments. All authors contributed to manuscript preparation.

## Competing interests

The authors declare no competing interests.
