## [Peer Review File · Nature Communications]

Spatial Transcriptomics Uncover Sucrose Post-phloem Transport During Maize Kernel DevelopmentREVIEWER COMMENTS

Reviewer #1 (Remarks to the Author):

This article utilizes the 10x Genomics Visium platform to construct a spatial transcriptome map of maize kernels from 12DAP to 24DAP. By leveraging this spatial transcriptome map, the authors have successfully identified marker genes specific to different cell types, thus providing a valuable data resource for future studies on maize kernel research.

Unfortunately, based on this dataset, only a limited number of cell types were detected, which was lower than expected, possibly due to the low resolution of 10X. What is valuable is that the authors have identified two key genes that are specifically expressed in the basal endosperm transfer layer (BETL), demonstrating their role in grain development and highlighting the significance of spatial transcriptome data in identifying key genes.

I have some suggestions aim to enhance the clarity and scientific value of the article, ensuring a more comprehensive and accurate representation of the research findings:

(1) The author mentions in the article that the average size of endosperm cells was 50 μm at 10 days after pollination (DAP) and 60 μm at 14 DAP. It should be noted that plant tissue cell sizes are not uniform, and the arrangement of spots on the 10x Genomics Visium platform may not completely align with the arrangement of cells (since the author does not provide a depiction of cell arrangement in the slices). Therefore, I suggest modifying this statement to avoid the misconception that each spot represents a single cell.

(2) The significance of the spatial transcriptome lies in annotating cell types using spatial information. Thus, I am eagerly anticipating the discovery of completely new cell types by the authors. In the article, the authors initially identified 25 cell types, which were later consolidated into 11 cell types through the integration of semi-thin sections and HE staining images. I am curious about the basis for this consolidation. Is there a possibility of discovering new cell types within the merged categories, and would this contribute to a deeper understanding of the grain's anatomical structure? In fact, I think the expression pattern of the BT2 gene in Figure 3c suggests the existence of additional cell types on the starchy endosperm (SE) and vascular endosperm (VE). Therefore, I recommend the authors carefully consider the rationale behind merging cell types.

(3) The authors constructed the spatial transcriptome map of the grain at 12, 18, and 24 days after pollination, but did not delve into further analysis regarding the trajectory of cell differentiation and development during these periods. This represents a missed opportunity. I suggest conducting a more detailed analysis and providing a comprehensive description of the developmental trajectory of key cell types.

(4) In Figure S10, the authors state that there are three major functional segments. However, the correlation between BETL and ESR appears to be relatively low, which is somewhat perplexing.

(5) The article showcases four slices, but most of the images do not indicate the name of the slice, making it difficult for readers to comprehend and determine the corresponding developmental period. Please ensure proper labeling of the slices.

(6) The presentation of Figure 1d could be improved. I recommend including the name of the Gene Ontology (GO) annotation, or at the very least, the abbreviation, on the image to

facilitate readability.

(7) What makes me more concerned is that nearly one-third of the 24DAP slices are missing, and 24DAP has no repetitions. Consequently, displaying the expression pattern of marker genes on this slice poses a risk. I suggest, at the very least, showcasing the effect of mRNA in situ capture by displaying the distribution map of unique molecular identifiers (UMIs) alone (without bright field photos) to confirm the absence of mRNA diffusion.

Reviewer #2 (Remarks to the Author):

The ms by Fu et al. on spatial transcriptomics of the maize kernel reports a valuable resource to the community by identifying markers for arguably the most important tissue of the crop, the edible kernel. The use of the Visium platform has its drawbacks in resolution, but I think, for the relatively large maize kernel, Visium is a fine tool at this point to identify markers specific to major cellular regions for the kernel. Figure 2 is the key evidence for validation of the methods and fidelity of the data. It was good to see a number of in situs done to validate the data. For the most part, the in situs and the Visium data agree. However, some of the presentation of that particular data was not entirely clear. In addition, some of the conclusions or statements in the ms do not seem justified. Here is more detail on those comments.

1. Visium resolution is 55 microns, but many of the cells in the kernel are smaller (i.e., non endosperm). The authors recognize this drawback, but there is no description of how this issue was handled in cells other than the endosperm in the analysis. I would imagine that some spots would have captured cells of two different identities at the transition between one cellular compartment vs. another. The authors need some more analysis on how resolution affects the transcriptome map they supply.
2. Figure 2 represents a good validation of the data with 12 in situ hybridizations of previously uncharacterized markers. First, I think the comparison of panels a-1 vs. m-x could be made more clear visually. It is hard for non-experts in kernel development to see where there is agreement in tissue localization. I would recommend showing the in situ and corresponding Visium profile together with better labeling.
3. The authors engage in a bit of circular logic or at least cherry picking in their explanation of the disagreement between in situs and the Visium profiles. If a,b,c,e,f,g,k,l agree well and this proves that the Visium data is accurate, then it is not clear how one could argue that cases of disagreement (d,h,i,j?) show the Visium profiling is more sensitive. The logical conclusion is that in cases of disagreement, it is not clear which is more sensitive or accurate.
4. I am not an expert on kernel anatomy and function, but the mutant characterization seems focused on gross morphology while the very specific localization of the sucrose transporters would imply a local function and a phenotype that reflects that local function?

Other minor comments:

5. It is not clear why 2,986 genes chosen to were used to construct the weighted gene

expression network. This comes without explanation in the ms.

6. It is not clear how the authors are defining “compartments.”

7. There are a lot of tissues to keep track of. I think the authors should organize and annotate the ms to better help the reader keep track of the anatomical regions to which they refer.

8. On the zmSUT1 7 being the result of duplication, presumably the whole sucrose transporter family was derived from duplication? I think the authors mean “recent” duplication.

9. The discussion of why Visium was more appropriate than single-cell RNA-seq or other spatial transcriptomic technologies is not really a good argument. Maize has been used for single-cell RNA-seq. Follow up in situ, as the authors performed, can provide validation in the absence of a large set of known markers. I don’t think the argument is needed. It seems the authors used Visium because it provided a convenient way to obtain many new markers for the kernel and localize them at the same time. The resolution is a tradeoff but the authors can address this, as per above.

10. If there is an online resource to explore this data, the authors should provide some guidance to it in the ms or the supplemental data.

REVIEWER COMMENTS

Reviewer #1 (Remarks to the Author):

This article utilizes the 10x Genomics Visium platform to construct a spatial transcriptome map of maize kernels from 12DAP to 24DAP. By leveraging this spatial transcriptome map, the authors have successfully identified marker genes specific to different cell types, thus providing a valuable data resource for future studies on maize kernel research. Unfortunately, based on this dataset, only a limited number of cell types were detected, which was lower than expected, possibly due to the low resolution of 10X. What is valuable is that the authors have identified two key genes that are specifically expressed in the basal endosperm transfer layer (BETL), demonstrating their role in grain development and highlighting the significance of spatial transcriptome data in identifying key genes.

Response: Thank you for your positive evaluation of our work. We appreciate your recognition of the importance and potential impact of our findings. We agree that this accomplishment holds great value as it establishes a valuable data resource for future studies in maize kernel research. By identifying these marker genes, we have contributed to the understanding of the molecular characteristics and functional diversity within maize kernels. This knowledge can be leveraged to explore various aspects of maize kernel development, metabolism, and regulatory mechanisms. We are grateful for your feedback and encouragement, and we hope that our research will contribute to the broader scientific community in understanding of maize kernels and their significance in agriculture and food production.

I have some suggestions aim to enhance the clarity and scientific value of the article, ensuring a more comprehensive and accurate representation of the research findings:

Response: Thank you for your valuable feedback. We took your comments seriously and carefully considered them in our revisions. In the following sections, we address your comments point by point.

(1) The author mentions in the article that the average size of endosperm cells was 50 μm at 10 days after pollination (DAP) and 60 μm at 14 DAP. It should be noted that plant tissue cell sizes are not uniform, and the arrangement of spots on the 10x Genomics Visium platform may not completely align with the arrangement of cells (since the author does not provide a depiction of

cell arrangement in the slices). Therefore, I suggest modifying this statement to avoid the misconception that each spot represents a single cell.

Response: Thank you for bringing this to our attention. We modified the statement to provide a clearer explanation that each spot represents a group or cluster of cells rather than a single cell. By making this adjustment, we aim to prevent any misunderstanding and provide a more precise representation of our research.

We have made the following modifications to the text:

“The tissue cells in maize kernels, such as the progeny embryo, endosperm, and maternal tissues, are of varying and uneven sizes. The average size of starchy endosperm cells at 10 DAP is 50 μm , and 60 μm at 14 DAP, whereas the cells in the embryo ranges from 10 to 30 μm . The diameter of the spots on the 10x Genomics Visium chip is 55 μm , resulting in instances where a single spot can encompass multiple cells. The aim of this technique is to identify molecular markers and perform cell clustering to investigate their biological functions. We expect that the same functional region consists of multiple spots that encompass cells. These spots, when within the same cluster, can be considered as biological replicates at the cellular level. This approach leads to more robust and reliable results, as it helps to minimize the impact of individual variations and increases the statistical power of the experiment (Fig. S3).”

(2) The significance of the spatial transcriptome lies in annotating cell types using spatial information. Thus, I am eagerly anticipating the discovery of completely new cell types by the authors. In the article, the authors initially identified 25 cell types, which were later consolidated into 11 cell types through the integration of semi-thin sections and HE staining images. I am curious about the basis for this consolidation. Is there a possibility of discovering new cell types within the merged categories, and would this contribute to a deeper understanding of the grain's anatomical structure? In fact, I think the expression pattern of the BT2 gene in Figure 3c suggests the existence of additional cell types on the starchy endosperm (SE) and vascular endosperm (VE). Therefore, I recommend the authors carefully consider the rationale behind merging cell types.

Response: Thank you for your valuable comments and questions regarding our article. As the reviewer mentioned, the significance of spatial transcriptomics lies in its capacity to aid in cell type annotation and enhance our understanding of cellular and tissue function, as well as organism development. Spatial transcriptomics enables the simultaneous identification of RNA in situ

expression data for approximately 40,000 genes across the entire maize genome. We sincerely value your anticipation regarding the discovery of entirely new cell types. To address your questions, we reanalyzed our data into Figure S7 and would like to provide the following explanation.

1) An additional figure illustrating the newly discovered cell types has been incorporated.

In our study, we initially identified 25 distinct cell types based on the similarity of gene expression data (Fig. 1b). It is noteworthy that more than half of these clusters are entirely novel findings. The discovery of these new cell types holds great potential for enhancing our understanding of the grain's anatomical structure and providing valuable insights into the diverse functions of cells in grain development, physiology, and quality. To ensure the preservation of information about the new clusters, we have included Figure S7 in the manuscript, showcasing the newly discovered cell populations. Notably, the figure illustrates that the pericarp is divided into three distinct cell types, the embryo into five groups, the starchy endosperm into five, and the vitreous endosperm into five. These categories align with the reviewer's observation that the expression pattern of the BT2 gene in Figure 3c indicates the presence of additional cell types in the endosperm.

2) We reserve 25 clusters for future data mining and 11 clusters for cross-comparison.

To facilitate cross-comparison with existing research and ensure the preservation of essential cellular diversity in the grain, we carefully merged the initial 25 clusters into 11 cell groups incorporating morphological data from semi-thin sections, HE staining images, and previous laser microdissection studies (Zhan, Plant Cell, 2015). We maintain both clustering approaches, allowing readers to choose the one that best suits their specific objectives. **The 25 cell populations provide valuable insights into understanding the function of unknown cell populations, while the 11 populations facilitate the comparison of gene expression differences across different research studies.**

3) We have made the following modifications to the text:

Spatial transcriptomics enables the simultaneous identification of RNA in situ expression data for approximately 40,000 genes throughout the entire maize genome. This high throughput capability enables heightened sensitivity to uncover novel cell types. Notably, the pericarp encompasses three distinct cell types, the embryo can be categorized into five new groups, the starchy endosperm can be classified into five groups, and the vitreous endosperm can be further divided into five groups (Fig. 1c and Fig. S7).

Figure S7. The defined 25 clusters after dimensional reduction.

The 25 clusters from dimensional reduction are mapped back to tissue sections, showing their location on the kernel. Each dot in the figure represents a tissue spot. The clusters are distinguished by different colors.

Reference:

Zhan J, *et al.* RNA sequencing of laser-capture microdissected compartments of the maize kernel identifies regulatory modules associated with endosperm cell differentiation. *Plant Cell* 27, 513-531 (2015).

(3) The authors constructed the spatial transcriptome map of the grain at 12, 18, and 24 days after pollination, but did not delve into further analysis regarding the trajectory of cell differentiation

and development during these periods. This represents a missed opportunity. I suggest conducting a more detailed analysis and providing a comprehensive description of the developmental trajectory of key cell types.

Response: We sincerely appreciate the reviewer for bringing this aspect to our attention. Pseudotime analysis is a computational method used to infer the temporal order of cellular states and gene expression changes during a biological process. It could be used to infer the alternative cell fates, oncogenic transformation and cell proliferation, especially to investigate the cell differentiation. In this project, our primary focus is on unraveling the mechanism of storage accumulation at filling stage (12- 24 DAP). It is known that maize kernels almost cease cell division and cell differentiation after 10 DAP. The cell fate is determined at the filling stages that our samples are unfit to run the pseudotime analysis to understand the cell trajectory. Actually, we are also very interested in the trajectory of cell differentiation that happened before 10 DAP. During the early stages, maize cell growth and development occur rapidly within a short period, exerting a significant influence on the fate of maize cells. **Our next project is currently underway to track changes in gene expression and reveal insights into cell fate determination and tissue development.** The tissue specific markers would be useful for tracking cell fate and differentiation of these cell types. Here, in the Discussion section, we have included an additional paragraph to provide a comprehensive description of the developmental trajectory of key cell types, thereby enhancing the readers' understanding of the dynamic nature of these cell types and their significance in the broader context of the study.

“Maize kernel development originates from double fertilization and mainly includes the cell division and differentiation (1-10 DAP), storage accumulation (10~35 DAP) and maturation stages (35~56 DAP). During the early stage, the fertilized ovule undergoes rapid cell division and differentiation, but significant storage compound accumulation hasn't occurred yet. After 10 DAP, cell fate is largely determined with the ceasing of cell differentiation, and the kernels start the accumulation of storage compounds such as starch granules, protein bodies, and oils. We look forward to future projects aimed at understanding cell trajectories, where it will be crucial to examine samples taken earlier than 10 DAP, when cell fate remains undetermined.”

(4) In Figure S10, the authors state that there are three major functional segments. However, the correlation between BETL and ESR appears to be relatively low, which is somewhat perplexing.

Response: It is true that the Pearson correlation between BETL and ESR appears to be relatively low. When we look closer, ESR is also lowly correlated with all other cell populations (< 0.07). The possibility is that ESR may have different biological functions and share less gene expression patterns with other cell types, leading to a weaker correlation between them. Very few studies have been conducted to study the function of ESR region, possibly due to its short appearance within the narrow window of 8-12 DAP. However, it has been proposed to play a crucial role in supporting the development and growth of the embryo. Based on our analysis, the differentially expressed genes in the ESR are enriched in embryo defense and signaling between the embryo and the endosperm. **Thus, we deleted such confusing statement “ESR was correlated with BETL” from the text.**

(5) The article showcases four slices, but most of the images do not indicate the name of the slice, making it difficult for readers to comprehend and determine the corresponding developmental period. Please ensure proper labeling of the slices.

Response: Thank you for your advice. We have reviewed all the figures and made the necessary modifications, labeling them as S12D_1, S18D_1, S18D_2, and S24D_1. Additionally, we have included a description in the Figure 1 legend stating that " S12D_1 indicates the sample taken at 12 DAP; S18D_1 indicates the sample taken at 18 DAP; S18D_2 indicates the biological sample taken at 18 DAP; S24D_1 indicates the sample taken at 24 DAP."

(6) The presentation of Figure 1d could be improved. I recommend including the name of the Gene Ontology (GO) annotation, or at the very least, the abbreviation, on the image to facilitate readability.

Response: Thank you so much for your valuable tips. We have replaced the numbers with the names of the GO terms, making it much easier to understand.

(7) What makes me more concerned is that nearly one-third of the 24DAP slices are missing, and 24DAP has no repetitions. Consequently, displaying the expression pattern of marker genes on this slice poses a risk. I suggest, at the very least, showcasing the effect of mRNA in situ capture by displaying the distribution map of unique molecular identifiers (UMIs) alone (without bright field photos) to confirm the absence of mRNA diffusion.

Response: We appreciate your suggestion. To address this issue, we have taken your feedback into consideration and made the necessary changes. As a result, we have included the effect of mRNA in situ capture by presenting the distribution map of unique molecular identifiers (UMIs) in Figure S5, which present the missing region of the 24DAP slice and further supports the accuracy of our results. Additionally, we have incorporated our explanation directly into the main text.

1) The technology is new and expensive.

This revolutionary technology of Visium Spatial Transcriptomics was introduced in 2020, but it currently carries a higher price tag about ~\$1000 for each chip. Considering the high cost, we did not replicate all time points biologically. Instead, we performed a biological replicate at the middle filling stage of 18 DAP to account for natural biological variability and to ensure the reliability and robustness of the results.

2) The sample of 24 DAP is abundant of starch at the late filling stage.

The published papers on spatial transcriptomic studies typically focus on plant meristems or leaves, which contain fewer storage metabolites and almost consistent tissue texture and density. In contrast, our samples were collected during the early, middle, and late filling stages (12, 18, and 24 DAP), specifically targeting storage accumulation. At the late filling stage of 24 DAP, maize kernels become harder, denser and brittle due to the accumulation of starch and other storage compounds. As the amount of starch increases as the kernel matures, more programmed cell death (PCD) takes place in the starchy endosperm. The distribution of different textures and density within the kernel, such as the endosperm, embryo, and pericarp make the tissue uneven, leading to the generation of fragmented or uneven sections during tissue sectioning. To overcome the technological challenges, our research group has collaborated with Shanghai Ouyi Company, whose technical personnel have extensive experience in animal rather than plant tissues. With their guidance, we still took two years to optimize the experimental procedure including kernels embedding, sectioning, and tissue permeabilization. However, it is challenging to obtain intact tissue sections from 24-DAP kernels due to their high starch content.

3) The remaining spots are enough to group cell types.

In Figure S5, we observed that the largest cell population of the central starchy endosperm of the 24-DAP was missing due to its high starch content, while the peripheral region retained a sufficient number of spots to allow for partitioning with the surrounding spots. Assuming that the region of starchy endosperm occupies 200 spots, even if 150 spots are lost, the remaining spots from the

same cell population can still effectively partition them since 200 spots can be considered biological replicates. To validate this, we conducted a correlation analysis between the residual spots in the starchy region from 24-DAP and the ones from 18-DAP. We found a strong and consistent correlation (Figure Q7), confirming the reliability of the data and confidence in the findings. We utilize Unique Molecular Identifiers (UMI) to map the quantities of RNA molecules onto the sample sections (Fig.S5b and d). Our findings indicate that the missing tissue corresponds to the starchy region, characterized by a high concentration of starch. This region is represented by the dark blue color, indicating low expression levels of genes. However, it is important to note that there are still remaining sections of similar tissue type, which can potentially compensate for the missing information (Fig.S5).

Figure Q7. Pearson correlation of 0.923 indicates the high consistency of spots in starchy endosperm between S18D_1 and S24D_1.

Figure S5. The density of expressed genes and transcripts in a spot.

Reviewer #2 (Remarks to the Author):

The ms by Fu et al. on spatial transcriptomics of the maize kernel reports a valuable resource to the community by identifying markers for arguably the most important tissue of the crop, the edible kernel. The use of the Visium platform has its drawbacks in resolution, but I think, for the relatively large maize kernel, Visium is a fine tool at this point to identify markers specific to major cellular regions for the kernel.

Response: Thank you for your valuable feedback on our manuscript. We appreciate your recognition of the study as a valuable resource for the community, particularly in identifying markers for the crucial tissue of the crop, the edible kernel. Considering the relatively large size of the maize kernel, we agree that Visium is currently a suitable tool for identifying markers specific to the major cellular regions within the kernel. We believe that the findings presented in the manuscript contribute significantly to our understanding of the spatial gene expression patterns in this important agricultural crop.

Figure 2 is the key evidence for validation of the methods and fidelity of the data. It was good to see a number of in situs done to validate the data. For the most part, the in situs and the Visium data agree. However, some of the presentation of that particular data was not entirely clear. In addition, some of the conclusions or statements in the ms do not seem justified. Here is more detail on those comments.

Response: We appreciate your thoughtful comments on the manuscript, particularly regarding Figure 2, which serves as key evidence for validating the methods and data fidelity. We agree that the inclusion of multiple in situ experiments to validate the data is commendable. Thus, we compared our study with the previous report using laser-capture microdissection (LCM) in Table S9. We thoroughly re-evaluated the statements and make necessary adjustments to ensure that our conclusions are well-supported by the data presented.

1. Visium resolution is 55 microns, but many of the cells in the kernel are smaller (i.e., non-endosperm). The authors recognize this drawback, but there is no description of how this issue was handled in cells other than the endosperm in the analysis. I would imagine that some spots would have captured cells of two different identities at the transition between one cellular compartment

vs. another. The authors need some more analysis on how resolution affects the transcriptome map they supply.

Response: We appreciate the reviewer's comment regarding the resolution of the Visium technology and its potential impact on the analysis. To address this concern, we have conducted data validation by comparing the previous study using laser-capture microdissection (Table S9) to support the accuracy of our results. We also added our explanation to address any potential uncertainties or limitations associated with the Visium technology's resolution. Still, this did not alter our primary objective of accurately identifying gene markers. We are grateful for the reviewer's valuable feedback, and we believe that the inclusion of these measures strengthens the robustness and credibility of our study. Here is the details:

1) The large kernel size at filling stage could offset limited spatial resolution from Visium technologies.

The primary technological limitation of these spatial gene expression platforms is resolution, with the unit of observation being spots that are 55 μm in diameter on the Visium platform. As the reviewer pointed out that the maize kernel is relatively large. The average diameter of an endosperm cell was 50-80 μm at kernel filling stage, allowing a single cell in each spot for endosperm tissues. Still, certain spots could capture 1-5 cells from the pericarp, aleurone, and embryo depending on the biological tissue. If the maize kernel is covered by 2500 spots and 25 cell types, there are 100 spots per region, which are equivalent to 100 biological replicates for mutual validation, ensuring high accuracy of cell clustering.

It is true that some spots would have captured cells of two different identities at the transition between one cellular compartment and another. In our study, the gene markers used to define cell types are defined as highly expressed in specific tissues but are expressed at low levels or not expressed in the remaining cell population. Even if two cell types are mixed within a single spot, it will not alter the marker status or cell clustering, but only result in a decrease in the value for that particular spot. Therefore, this does not alter our objective of identifying molecular markers and cell identity.

2) Marker validation using laser-capture microdissection

To further validate the markers we have identified, we conducted a comparison with a previous study that utilized laser-capture microdissection (LCM) to profile mRNA populations within the

primary cell types. By comparing their markers with ours, we observed a consistent overlap, indicating the accuracy and reliability of our methodology (Table S9).

Table S9. The comparison of defined markers between LCM and Spatial transcriptome.

Gene ID	ST													LCM		Gene function
	AL	BETL	CZ	EAS	ESR	PC	PE	EM	SCU	SE	VE	max	Population	Highest CS score	Compartment	
Zm00001d012572	62.28	3.69	3.4	0.18	44.28	10.93	10.12	0.84	0.7	1.01	3.3	62.28	AL	0.51	AL	aleurone9
Zm00001d015569	4.74	0.22	0.6	0.53	0.78	0.94	0.72	0.45	0.51	0.13	0.33	4.74	AL	0.70	AL	vacuolar H ⁺ -translocating inorganic pyrophosphatase
Zm00001d000161	0.19	47.23	2.84	0.4	0.33	4.33	0.17	0.08	0.23	0.08	0.07	47.23	BETL	0.97	BETL	basal endosperm transfer layer 4 (BETL4)
Zm00001d003776	0.08	30.57	0.75	0.18	1.56	3.08	0.15	0.02	0.08	0.05	0.04	30.57	BETL	0.95	BETL	sucrose degrading enzyme (CWIN2, mm1)
Zm00001d009292	0.41	1	6.3	2.4	0.17	0.43	0.64	0.07	0.09	3.21	5	6.3	CZ	0.70	CZ	PLATZ transcription factor (floury3)
Zm00001d021289	1.77	2.77	9.47	3.96	0.28	0.83	1.66	0.19	0.19	5.71	9.06	9.47	CZ	0.70	CZ	late embryonic protein
Zm00001d002705	0.2	0.25	0.45	2.56	0.06	0.13	0.11	0.01	0.11	0.04	0.28	2.56	EAS	0.65	CSE (V&andSE)	ATP dependent copper transporter
Zm00001d015914	0.16	0.08	0.22	1.86	0	0.33	0.2	0.25	0.37	0.03	0.04	1.86	EAS	0.62	EM (EM&andSCU)	sugars will eventually be exported transporter4b
Zm00001d037985	0.05	0.26	0.74	1.3	0	0.55	4.98	29.36	5.36	0.61	0.34	29.36	EM	0.94	EM (EM&andSCU)	late EMryonic protein; EMryo specific protein5
Zm00001d043049	0.03	0.17	0.6	2.12	1.22	0.31	4.5	19.05	2.64	0.34	0.2	19.05	EM	0.81	EM (EM&andSCU)	non-specific lipid-transfer protein
Zm00001d027819	0.02	5.94	1.22	0.07	31.44	0.59	0.2	0.04	0.04	0.01	0.01	31.44	ESR	0.99	ESR	emryo surrounding region 2
Zm00001d053112	0.06	7.56	1.19	0.07	13.61	1.47	0.14	0.03	0.06	0.02	0.02	13.61	ESR	0.99	ESR	emryo surrounding region 6
Zm00001d025373	0	1.21	0.06	0.01	0	1.79	0.02	0	0.02	0	0	1.79	PC	0.32	PC	aluminum-activated malate transporter 8
Zm00001d048611	0.78	0.8	0.12	0.02	0.06	7.86	4.56	0.06	0.02	0.38	0.06	7.86	PC	0.85	PC	metallothionein-like protein 1B
Zm00001d020583	0.2	0.03	0.02	0.03	0	0.39	1.01	0.03	0.04	0.02	0.02	1.01	PE	0.95	PE	galactosyltransferase family protein
Zm00001d038476	0.15	0.02	0.02	0.02	0	0.11	1.25	0.01	0.01	0.05	0.03	1.25	PE	0.89	PE	alpha expansin5
Zm00001d026317	0.03	0.35	0.5	1.58	1.67	0.43	0.45	1.62	5.25	0.18	0.14	5.25	SCU	0.94	EM (EM and SCU)	Transcription factor PIF4
Zm00001d019504	1.02	0.26	1.15	3.64	8.56	1.26	2.18	7.47	10.92	0.77	0.42	10.92	SCU	0.91	EM (EM and SCU)	plasma membrane associated protein;tonoplast intrinsic proteins
Zm00001d045042	9.82	0.72	7.85	3.04	0.11	2.21	8.02	1.59	0.64	67.57	36.04	67.57	SE	0.89	CSE (V&andSE)	sucrose synthase 1 (sh1)
Zm00001d050032	1.77	0.2	1.34	0.47	0.44	0.79	2.12	0.29	0.11	19.84	10.4	19.84	SE	0.88	CSE (V&andSE)	glucose-1-phosphate adenyllyltransferase (bt2)
Zm00001d020592	454.82	15.17	191.28	153.4	3.22	12.36	198.52	9.51	19.3	505.1	771.71	771.71	VE	0.96	CSE (V&andSE)	27-kD y-zein
Zm00001d005793	268.27	7.19	80.4	162.01	2.06	7.6	120.84	6.38	16.24	366.71	483	483	VE	0.95	CSE (V&andSE)	16-kD y-zein (mucronate1)

2. Figure 2 represents a good validation of the data with 12 in situ hybridizations of previously uncharacterized markers. First, I think the comparison of panels a-1 vs. m-x could be made clearer visually. It is hard for non-experts in kernel development to see where there is agreement in tissue localization. I would recommend showing the in situ and corresponding Visium profile together with better labeling.

Response: In regard to the issue you mentioned, we have made modifications to the images. We have placed the experimental in situ image alongside the control image, while also including temporal annotations. To facilitate a visual demonstration of the consistency in tissue localization between in situ and spatial transcriptomics, **we have outlined the localized regions with dashed lines for reader convenience.**

Figure 2. Marker gene validation using experimental and electronical RNA in situ hybridization.

3. The authors engage in a bit of circular logic or at least cherry picking in their explanation of the disagreement between in situs and the Visium profiles. If a,b,c,e,f,g,k,l agree well and this proves that the Visium data is accurate, then it is not clear how one could argue that cases of disagreement (d,h,i,j?) show the Visium profiling is more sensitive. The logical conclusion is that in cases of disagreement, it is not clear which is more sensitive or accurate.

Response:

We appreciate the reviewer's comment regarding the perception of selective reasoning. In the revised version of the manuscript, we have provided an explanation why this discrepancy could happen. Additionally, we would like to inform you that in the new version, the labels of Fig. 2d, 2h, 2i, and 2j have been updated to Fig. 2g, 2o, 2q, and 2s, respectively. Here is the detail:

1) Off-target binding or cross-hybridization for RNA in situ hybridization.

While RNA in situ hybridization (RNA-ISH) is a valuable technique for studying gene expression and RNA localization for one gene per experiment, it does have some limitations and disadvantages. It is primarily a qualitative technique, providing information about the presence and localization of RNA molecules rather than precise quantification compared to other methods such as quantitative PCR or RNA sequencing. RNA probes used in RNA-ISH should ideally be highly specific to the target RNA sequence to avoid cross-reactivity with other RNA molecules. However, achieving complete specificity can be challenging, and there is always a possibility of off-target binding or cross-hybridization. The non-specific binding of RNA probes or background staining can result in false-positive signals, leading to inaccurate interpretation of results.

Conducting RNA-ISH in seeds may present some challenges due to the presence of complex seed structures, such as the seed coat, endosperm, and embryo. It is important to note that not all genes are suitable for validation through in situ hybridization. While some specific genes exhibit stability within the system, others may experience non-specific signal binding during the hybridization process. For instance, in a study conducted by Liu X (2014), the genes of *zm.13387*, *zm.2941* and *zm.105* were found to be embryo-specific based on microarray expression profiles and RT-PCR analyses. Nevertheless, the results revealed different hybridization signals present in both the endosperm and embryo.

Reference:

Liu XQ, et al. Identification and characterization of promoters specifically and strongly expressed in maize embryos. *Plant Biotechnology Journal*. 12, 1286-1296 (2014).

2) The spatial transcriptomics is more sensitive in measuring gene expression than RNA in situ hybridization

Spatial transcriptomics and experimental RNA in situ hybridization are two methods for studying gene expression, but they have different strengths and limitations. Spatial transcriptomics is a high-throughput sequencing method in capturing the relative abundance and spatial distribution of RNA

molecules within tissues. The ST becomes a highly accurate and unbiased method of transcriptome measurement especially with the development of second-generation sequencing technologies.

When we look closer at the expression data from spatial transcriptomics (See Table Q3 below), we discovered that the genes are not exclusively expressed in a single cell type. Instead, they exhibit low expression levels in neighboring tissues as well. For instance, as shown in Figure 2o, gene Zm00001d043049 is primarily expressed in EM (19.05), but it also exhibits low expression in SCU (2.64). This discrepancy can be distinguished through the quantification provided by spatial transcriptomics, whereas RNA in situ hybridization may produce signals in both cell types. This observation emphasizes the heightened sensitivity of quantitative analysis (spatial transcriptomics) compared to qualitative analysis (RNA in situ hybridization).

Table Q3. The gene expression using Spatial Transcriptomics

GeneID	Figure 2	cell type 1	cell type 2
Zm00001d018727	Figure 2g	SE(18.93)	VE(5.55)
Zm00001d043049	Figure 2o	EM(19.05)	SCU(2.64)
Zm00001d052136	Figure 2q	BETL(1.5)	PC(17.27)
Zm00001d038558	Figure 2s	BETL(1.75)	PC(9.07)

4. I am not an expert on kernel anatomy and function, but the mutant characterization seems focused on gross morphology while the very specific localization of the sucrose transporters would imply a local function and a phenotype that reflects that local function?

Response: Thank you for your comment. You raise an important point regarding the focus on gross morphology in the mutant characterization compared to the specific localization of sucrose transporters and their implication for local function and phenotype.

The basal endosperm transfer layer (BETL) in maize refers to a specialized tissue layer located at the base of the endosperm, which is the nutritive tissue surrounding the embryo in the seed. BETL plays a crucial role in nutrient transport between the maternal tissues (such as the placenta) and the developing kernel. It ensures the efficient uptake and distribution of nutrients, which are necessary for embryo development and seed maturation. By understanding the local function of the BETL and its impact on overall grain development, researchers can gain insights into the intricate processes involved in nutrient transport and optimize crop yield and quality. When the genes of *ZmSUT* were silenced, we found that the cell wall ingrowth (CWI) in the BETL region of *zmsut1/7RNAi* seeds was apparently inhibited, indicating *ZmSUT1* and *ZmSUT7* are essential

for CWI formation (Fig. 4b). The kernel size and weight of the self-pollinated *zmsut1/7*RNAi transgenic lines were greatly reduced compared with the control (Fig. 4d and 4e). **Therefore, mutations occurring in specific genes in the BETL region not only affect the phenotype of that particular area in the grain, but also lead to issues in the overall phenotype of the grain due to disruptions in nutrient transport.**

Figure 4. Functional study of *ZmSUT* genes in maize seed.

Other minor comments:

5. It is not clear why 2,986 genes chosen to were used to construct the weighted gene expression network. This comes without explanation in the ms.

Response: We apologize for the confusion. Here is the revised sentence for the main text:

"To construct the gene co-expression network, we extracted 2,986 genes from the initial pool of 26,161 genes by excluding those with low expression levels (< 1 UMI) or low variability (coefficient of variation < 0.8) across different cell types that usually represent noise. Subsequently, we performed a weighted gene correlation network analysis (WGCNA) and identified eleven co-expression modules (Table S5 and Fig. S9a)."

And here is the sentence to be added to the methods section:

"The dataset comprising 26,161 genes underwent filtration as genes with low expression levels or lack of variability are typically considered as noise when constructing the gene co-expression network. The threshold for the minimum gene expression level was set at 0.3, while the coefficient of variation was set at 0.8. The gene co-expression network was established using the R package of WGCNA. We selected a soft-thresholding power of eight and a minimum gene number of 80 to identify highly correlated genes. The gene connectivity GS was determined for each module."

6. It is not clear how the authors are defining "compartments."

Response: We should have provided clearer explanations. Through dimensional reduction using spatial transcriptomic data, we identified 25 clusters that displayed significant similarities in gene expression. To enhance their biological significance, we incorporated anatomical information derived from semi-thin sections and HE dyeing images. Through the consolidation of these clusters, we were able to delineate eleven distinct functional cell populations. Notably, these cell populations, located within their respective compartments, exhibit shared biological functions due to their closely correlated gene expression and spatial proximity to one another.

Therefore, we included an explanation in the main text: In this context, the term "cell populations" is also referred to as "compartments," representing groups of cells with similar gene expression patterns and physical proximity, indicating their similar functions.

7. There are a lot of tissues to keep track of. I think the authors should organize and annotate the ms to better help the reader keep track of the anatomical regions to which they refer.

Response: I sincerely appreciate your invaluable guidance. It is indeed true that an abundance of functional compartments can easily perplex readers. Therefore, **we have introduced the utilization of prefixes to aid their comprehension and facilitate contextual tracking.** This includes: 1) referring to two regions derived from the maternal source as maternal-PC and maternal-PE; 2) denoting two compartments associated with the embryo as embryo-SCU and embryo-EM; 3) categorizing the seven endosperm compartments as follows: endosperm-BETL, endosperm-CZ, endosperm-EAS, endosperm-SE, endosperm-VE, endosperm-AL, and endosperm-ESR.

8. On the zmSUT1 7 being the result of duplication, presumably the whole sucrose transporter family was derived from duplication? I think the authors mean “recent” duplication.

Response: I agree with you that ZmSUT1 and ZmSUT7 were derived from a recent duplication based on the similarity of 93% in protein sequences and 91% in cDNA coding sequences. Consequently, we have made the necessary amendment to the sentence.

9. The discussion of why Visium was more appropriate than single-cell RNA-seq or other spatial transcriptomic technologies is not really a good argument. Maize has been used for single-cell RNA-seq. Follow up in situ, as the authors performed, can provide validation in the absence of a large set of known markers. I don't think the argument is needed. It seems the authors used Visium because it provided a convenient way to obtain many new markers for the kernel and localize them at the same time. The resolution is a tradeoff but the authors can address this, as per above.

Response: Thank you for your feedback. It is right that single-cell RNA sequencing and spatial transcriptomics are complementary techniques used to study gene expression and cellular heterogeneity. In our project, we opted for spatial transcriptomic technologies, specifically Visium, due to its capability to simultaneously explore multiple novel markers. However, it is important to note that this choice comes with a tradeoff in resolution compared to single-cell RNA-seq, which provides higher-resolution gene expression data at the single-cell level. **Consequently, we have removed the argument regarding single-cell RNA-seq from the discussion section.**

10. If there is an online resource to explore this data, the authors should provide some guidance to it in the ms or the supplemental data.

Response: Thank you for your feedback. **We have incorporated the methods for exploring the online resource and visualizing electronic RNA in situ hybridization into the supplementary file as following.**

“The website (<http://119.78.67.206:3838/>) is dedicated to visualizing electronic RNA in situ hybridization images of the tissue sections from the maize inbred line W64A. On the main interface of the webpage, you will find a demonstration examples that utilizes known marker gene.

To retrieve specific electronic in situ images, simply enter a gene ID (e.g., Zm00001d012572) into the search box. After submitting the gene ID, it may take approximately 10 seconds to 2 minutes,

depending on your internet connection, to retrieve the results. You will receive four electronic in situ images corresponding to different developmental stages of the maize kernel.

The "slice1" image represents the maize kernel at 12 days after pollination (DAP), while "slice1.1" and "slice1.2" are two biological replicates, representing the maize kernel at 18 DAP. Additionally, "slice1.3" represents the maize kernel at 24 DAP.

Please note that the gene ID used is based on the B73v4 reference genome. If you have gene IDs from other versions, please visit maizeGDB to convert them accordingly.”

REVIEWER COMMENTS

Reviewer #1 (Remarks to the Author):

The authors have addressed most of my concerns. There are a few remaining concerns, which I hope can be addressed to improve the manuscript.

(1) Please offer more details of methods which would help readers to follow and validate, especially about the cell type classification method. Which parameters were used in Seurat analysis pipeline?

(2) As an important result, the proposed new cell types were not solid. As the author mentioned in the article, they “found that the pericarp encompasses three distinct cell types, the embryo can be categorized into five new groups, the starchy endosperm can be classified into five groups, and the vitreous endosperm can be further divided into five groups”. However, are there enough marker genes (including novel marker gene and well-characterized genes from the literature) and in situ hybridization experiments to support these new findings? What is important if the completely new cell types that can be validated by marker genes?

(3) The author mentioned: “We discovered that maternal-PE was closer to endosperm compartments, and that maternal-PC was more correlated with endosperm-BETL, suggesting that the gene expression programs in maternal and filial tissues are subjected to convergent evolution, reaching adapted functions in spite of their divergently genetic origins.” I think the convergent evolution is inappropriate here. Convergent evolution refers to the phenomenon that two or more distantly related organisms evolve into similar morphological features or structures due to living in the same type of environment, the author should carefully consider this statement to avoid ambiguity.

(4) The co-expression analysis is too simple here, it seems to be just for a GO analysis, where the author can show some cell type-specific regulatory networks, which can further improve the credibility of the GO result. And I'm not very clear if the authors did multiple test correction when they did Go enrichment analysis.

(5) The logic between different sections is somewhat confusing. For example, in the section "Mining key genes essential for sucrose transport and storage accumulation", the author proposed "These results together confirmed that the technology of spatial transcriptomics is more sensitive and robust than the experimental RNA in situ hybridization", I think this conclusion should be put forward when the author introduces the data quality of 10x Genomics Visium to emphasize the technical advantages of spatial transcriptome.

Reviewer #2 (Remarks to the Author):

The authors have addressed my comments and I their responses and changes to the manuscript satisfy my concerns.

REVIEWER COMMENTS

Reviewer #1 (Remarks to the Author).

The authors have addressed most of my concerns. There are a few remaining concerns, which I hope can be addressed to improve the manuscript

Thank you for your positive response to the addressed concerns. We greatly appreciate your valuable input, as it plays a crucial role in enhancing the quality and clarity of our study. We have carefully considered your comments regarding the methods, data reliability, and further network analysis, and have made significant improvements to the manuscript accordingly. We believe that these revisions have significantly enhanced the overall quality of the study. Thank you once again for your valuable feedback.

(1) Please offer more details of methods which would help readers to follow and validate, especially about the cell type classification method. Which parameters were used in Seurat analysis pipeline?

Thank you for your valuable suggestions. In order to provide more details for readers to follow, we included additional information about the methods used, particularly regarding the cell type classification method. In our study, we utilized the Seurat R package developed and maintained by the Satija lab (<https://satijalab.org/seurat/>) to define cell types. Seurat has gained widespread popularity in the field of spatial transcriptomics due to its robustness, scalability, and user-friendly interface. It offers various modules that facilitate the analysis workflow, enabling researchers to perform tasks such as quality control, normalization, dimensionality reduction, clustering, and visualization.

In our analysis, we leveraged specific Seurat modules. We used the FindNeighbors function to calculate neighbor cell relationships based on gene expression patterns. This step helped identify cells with similar gene expression profiles, which is crucial for subsequent clustering analysis. We then applied the FindClusters function to cluster the cell populations, grouping together cells with similar expression patterns.

To improve the reproducibility of our study and encourage further exploration of our methods, we have uploaded our analysis scripts to the GitHub website (<https://github.com/wwq413/SpatialTranscriptomics>). Additionally, we have included the relevant parameters in our Methods section and cited the appropriate reference accordingly. This transparency and accessibility aim to facilitate the reproducibility of our findings and encourage further investigations in the field.

(2) As an important result, the proposed new cell types were not solid. As the author mentioned in the article, they "found that the pericarp encompasses three distinct cell types, the embryo can be categorized into five new groups, the starchy endosperm can be classified into five groups, and the vitreous endosperm can be further divided into five groups". However, are there enough marker genes (including novel marker gene and we -characterized genes from the literature) and in situ hybridization experiments

to support these new findings? What is important if the completely new cell types that can be validated by marker genes?

Thank you for your valuable feedback. We acknowledge your concern regarding the validation of the proposed new cell types in our study. While we have identified distinct cell types in the pericarp, embryo, starchy endosperm, and vitreous endosperm, we understand that it is crucial to provide sufficient evidence to support these findings.

In response to your comment, we have performed additional analyses to validate for the identified 25 clusters. We have expanded our investigation by including a comprehensive set of marker genes, including both novel markers discovered in our study and well-characterized genes from the existing literature, including RNA-seq from manual dissection or from laser microdissection, and experimental in situ hybridization. By incorporating these additional analyses and experiments, we have strengthened the validity of our findings and reliability of our results.

Additionally, we have included Fig. S7 and Table S3 into our manuscript to support our conclusions. We can see that almost all regions could be successfully validated using alternative technologies. To demonstrate the presence of the newly defined cell populations, we present some examples here. For instance, the marker genes of Zm00001d033447 defined by spatial transcriptomics (Figure 1: left panel) are consistent with the results obtained from RNA-seq analysis of manually dissected samples (Figure 1: right panel) (Doll et al., 2020, *The Plant Cell*). Similarly, the marker genes of Zm00001d018254 defined by spatial transcriptomics (Figure 2: left panel) show agreement with the results from RNA-seq analysis of samples obtained through laser microdissection (Figure 2: right panel) (Zhan et al. 2015, *The Plant Cell*). Furthermore, the marker genes of Zm00001d050577 defined by spatial transcriptomics (Figure 3: left panel) align with the results obtained from in situ hybridization experiments (Figure 3: right panel) (Doll et al., 2020, *The Plant Cell*).

However, it is worth noting that some small cell populations are challenging to validate using current technologies, despite multiple attempts we tried using in situ hybridization experiments. The signals in these cases may be overlooked by human observers due to the small size of the areas under investigation. For example, the width of cluster c18 or c23 is approximately 110 μm . As mentioned in other studies, achieving such specificity in in situ hybridization experiments can be difficult, and non-specific adsorption signals cannot be completely avoided.

Cluster ID	Marker genes from Spatial transcriptomics	Marker genes from RNA-seq using manual or laser laser microdissection	Marker genes using experimental in situ hybridization
C0	Zm00001d023955, Zm00001d052136, Zm00001d027854	Zm00001d006669	Zm00001d027854, Zm00001d052136
C1	Zm00001d008925, Zm00001d027291, Zm00001d017438	Zm00001d008925, Zm00001d027291, Zm00001d017438	-
C2	Zm00001d037382, Zm00001d015515, Zm00001d024522	Zm00001d037382, Zm00001d015515, Zm00001d024522	-
C3	Zm00001d024996, Zm00001d042541, Zm00001d018629	Zm00001d024996, Zm00001d042541, Zm00001d018629	-
C4	Zm00001d046596	Zm00001d046596	-
C5	Zm00001d046126	Zm00001d046126	Zm00001d046126
C6	Zm00001d020395, Zm00001d033447	Zm00001d020395, Zm00001d033447	-
C7	Zm00001d049179, Zm00001d048643	Zm00001d049179, Zm00001d048643	-
C8	Zm00001d033905, Zm00001d041489, Zm00001d035439	Zm00001d033905, Zm00001d041489	Zm00001d035439
C9	Zm00001d041822, Zm00001d052759, Zm00001d019277	Zm00001d041822, Zm00001d052759	Zm00001d041822
C10	Zm00001d007299, Zm00001d012572, Zm00001d020938	Zm00001d012572, Zm00001d020938	Zm00001d046599, Zm00001d012572
C11	Zm00001d050577, Zm00001d017285, Zm00001d037439	Zm00001d050577, Zm00001d017285, Zm00001d037439	Zm00001d050577, Zm00001d017285, Zm00001d037439
C12	Zm00001d009292, Zm00001d013159	Zm00001d009292, Zm00001d013159	Zm00001d009292, Zm00001d013159
C13	Zm00001d003677, Zm00001d013159	Zm00001d003677, Zm00001d013159	Zm00001d013159
C14	Zm00001d029696, ENSRNA049478426	-	Zm00001d037498
C15	Zm00001d051653, Zm00001d033714, Zm00001d043610	-	Zm00001d009646
C17	Zm00001d028714	Zm00001d028714	-
C16	Zm00001d018727	Zm00001d018727	-
C18	Zm00001d013956, Zm00001d022464, Zm00001d040127	-	-
C19	Zm00001d048808	Zm00001d048808	Zm00001d048808
C20	Zm00001d020591	Zm00001d020591	Zm00001d020591
C21	Zm00001d030855, Zm00001d035760	Zm00001d030855, Zm00001d035760	Zm00001d030855, Zm00001d035760
C22	Zm00001d045937	Zm00001d045937	Zm00001d045937
C23	Zm00001d025343	-	-
C24	Zm00001d027819, Zm00001d020780, Zm00001d011342	Zm00001d027819, Zm00001d020780, Zm00001d011342	Zm00001d027819

Table S3 in manuscript. The summary of marker genes for 25 clusters from Spatial transcriptomics and the literature including RNA-seq from manual dissection or laser microdissection, as well as experimental in situ hybridization.

Figure S7 in manuscript. Representatives of marker genes defined by spatial transcriptomic data in the tissues of pericarp, embryo and endosperm.

The left panel (e.g., a-PE) provides an overview of the 25 cell populations, while the right panel (e.g., a-PE) displays a snapshot of the spatial expression of marker genes. The abbreviations PE, SCU, EM, SE, and VE correspond to pericarp, scutellum, embryo meristem, starchy endosperm, and vitreous endosperm, respectively.

Figure 1. The markers genes of Zm00001d033447 defined by spatial transcriptomics is consistent with the results of RNA-seq from manual dissection.

Figure 2. The markers genes of Zm00001d018254 defined by spatial transcriptomics is consistent with the results of RNA-seq from laser microdissection.

Figure 3. The markers genes of Zm00001d050577 defined by spatial transcriptomics is consistent with the results of experiment in situ hybridization.

(3) The author mentioned: "We discovered that maternal-PE was closer to endosperm compartments, and that maternal-PC was more correlated with endosperm-BETL suggesting that the gene expression programs in maternal and filial tissues are subjected to convergent evolution, reaching adapted functions in spite of their divergently genetic origins." I think the convergent evolution is inappropriate here. Convergent evolution refers to the phenomenon that two or more distantly related organisms evolve into similar morphological features or structures due to living in the same type of environment, the author should carefully consider this statement to avoid ambiguity.

Thank you for your valuable feedback. We appreciate your input regarding the use of

the term "convergent evolution" in our statement. Upon reconsideration, we agree that the term may not be the most appropriate choice in this context. Instead, we revised the statement to emphasize the cooperative nature of gene expression programs in maternal and filial tissues, despite their divergent genetic origins. "We discovered that maternal-PE was closer to the endosperm compartments, while maternal-PC showed a stronger correlation with endosperm-BETL. This suggests that the gene expression programs in maternal and filial tissues are cooperatively functioning, despite their divergent genetic origins."

(4) The co-expression analysis is too simple here, it seems to be just for a GO analysis, where the author can show some cell type-specific regulatory networks, which can further improve the credibility of the GO result. And I'm not very clear if the authors did multiple test correction when they did Go enrichment analysis.

Thank you for your valuable feedback. We appreciate your suggestion to enhance the credibility of our GO analysis by incorporating cell type-specific regulatory networks. To address this, we performed Weighted correlation network analysis (WGCNA) to identify modules of highly correlated genes and construct functional networks. Through correlation analysis, we found that each module was closely associated with eleven corresponding anatomical regions. Furthermore, the GO functional annotations of these modules revealed enrichment in specific biological processes, including starch biosynthetic process, fatty acid biosynthetic process, and protein reservoir activity. These findings are consistent with our samples collected from the maize filling stage, during which storage accumulation is highly active. Upon closer examination of our gene expression dataset, we observed that the enriched genes involved in lipid, starch, and protein pathways exhibited high expression levels (Table S9) and belonged to "star genes" that have been previously studied in maize. These include gene families involved in such as oleosin in the embryo, starch synthesis in starchy endosperm, and zein biosynthesis in vitreous endosperm. These findings provide strong validation for the reliability of our GO enrichment analysis. We thus incorporated this information into our manuscript (Figure 2) to provide a more comprehensive understanding of the gene expression patterns and their functional implications.

During the GO enrichment analysis, we performed multiple test correction to account for the potential issue of false positives. Specifically, we utilized established methods such as the Bonferroni correction or the Benjamini-Hochberg procedure to adjust the p-values obtained from the enrichment analysis. This rigorous statistical analysis strengthens the reliability of the GO results and ensures a more robust interpretation of our findings. We have included this information in the revised manuscript to provide transparency and address any potential concerns.

GeneID	EM	SCU	SE	VE	max	sum	Population	Gene function	Pathway
Zm00001d002768	18.0	38.7	1.9	1.8	38.7	119.8	SCU	oleosin 1	lipid
Zm00001d045988	15.5	39.8	3.5	1.9	39.8	103.0	SCU	oleosin 2	lipid
Zm00001d051459	11.2	17.6	1.6	1.0	17.6	58.6	SCU	oil body-associated protein 1B	lipid
Zm00001d043464	9.9	18.2	1.6	0.9	18.2	56.3	SCU	oil body-associated protein 2A	lipid
Zm00001d053228	9.2	11.8	1.2	0.9	11.8	44.4	SCU	acyl carrier protein (ACP)	lipid
Zm00001d033612	6.6	16.3	1.0	0.6	16.3	43.6	SCU	oleosin 4	lipid
Zm00001d032019	5.0	8.0	1.5	0.9	8.0	31.3	SCU	acyl carrier protein 1	lipid
Zm00001d003938	5.4	10.4	1.0	0.5	10.4	28.8	SCU	steroleosin	lipid
Zm00001d042344	2.6	6.9	2.6	1.4	6.9	25.5	SCU	stearoyl-ACP desaturase1	lipid
Zm00001d011755	5.1	6.6	0.7	0.3	6.6	22.0	SCU	oil body-associated protein 2B	lipid
Zm00001d022616	3.4	6.1	0.7	0.6	6.1	20.5	SCU	acyl carrier protein 2 (ACP)	lipid
Zm00001d006080	4.3	6.0	0.5	0.3	6.0	18.9	SCU	oleoyl-ACP hydrolase	lipid
Zm00001d021046	2.7	6.3	1.2	0.6	6.3	18.9	SCU	oleoyl-ACP thioesterase A	lipid
Zm00001d004125	3.3	4.9	0.8	0.5	4.9	18.8	SCU	acetyl-CoA carboxylase 2	lipid
Zm00001d045042	1.6	0.6	67.6	36.0	67.6	137.6	SE	sucrose synthase 1 (sh1)	starch
Zm00001d044129	0.4	0.3	29.0	17.6	29.0	69.6	SE	glucose-1-phosphate adenylyltransferase (sh2)	starch
Zm00001d050032	0.3	0.1	19.8	10.4	19.8	37.8	SE	glucose-1-phosphate adenylyltransferase (bt2)	starch
Zm00001d015746	0.2	0.1	14.0	8.6	14.0	29.8	SE	adenine nucleotide transporter (bt1)	starch
Zm00001d039512	0.5	0.2	5.4	2.2	5.4	14.7	SE	hexokinase1	starch
Zm00001d045261	0.6	0.6	4.4	1.7	4.4	10.7	SE	starch synthase 1 (ss1)	starch
Zm00001d000002	0.1	0.2	4.0	1.4	4.0	9.1	SE	dull endosperm1; starch synthase 2 (du1,ss2)	starch
Zm00001d037234	0.2	0.2	1.7	1.3	1.7	5.8	SE	sugary2 (su2); starch synthase IIa (ss)	starch
Zm00001d004438	0.2	0.1	1.6	1.0	1.6	5.3	SE	starch debranching enzyme1 (dbe1)	starch
Zm00001d020592	9.5	19.3	505.1	771.7	771.7	2334.4	VE	27-kD γ -zein	protein
Zm00001d035760	7.6	16.1	285.6	570.7	570.7	1780.7	VE	15-kD β -zein	protein
Zm00001d005793	6.4	16.2	366.7	483.0	483.0	1520.7	VE	16-kD γ -zein (mucronate1)	protein
Zm00001d019155	5.5	7.8	256.8	498.6	498.6	1513.9	VE	19-kD α -zein	protein
Zm00001d049243	4.2	7.3	132.5	277.6	277.6	860.7	VE	22-kD zein (floury2)	protein
Zm00001d045937	2.0	1.7	49.2	266.0	266.0	650.1	VE	10-kD -zein	protein
Zm00001d048848	1.8	2.2	110.0	214.7	214.7	569.4	VE	19-kD zein	protein
Zm00001d049476	1.8	1.8	74.6	152.1	152.1	407.8	VE	19-kD zein	protein
Zm00001d048813	0.8	1.2	29.1	102.4	102.4	265.4	VE	19-kD zein	protein
Zm00001d030855	1.0	1.3	46.8	95.1	95.1	248.6	VE	19-kD zein	protein
Zm00001d048816	0.7	1.3	15.2	72.5	72.5	202.6	VE	22-kD zein	protein
Zm00001d048850	0.4	0.5	30.1	64.4	64.4	174.1	VE	19-kD zein	protein
Zm00001d048812	0.5	0.5	16.8	56.9	56.9	149.1	VE	22-kD zein	protein
Zm00001d048847	0.4	0.3	25.1	51.4	51.4	134.2	VE	19-kD zein	protein
Zm00001d035700	0.4	0.4	12.3	36.4	36.4	117.8	VE	legumin 1; 51-kD globulin	protein
Zm00001d048809	0.4	0.3	9.2	32.7	32.7	87.2	VE	22-kD zein	protein
Zm00001d048849	0.2	0.1	6.2	16.3	16.3	43.9	VE	19-kD zein (floury4)	protein
Zm00001d048818	0.1	0.0	3.2	7.9	7.9	21.8	VE	22-kD zein	protein
Zm00001d038597	0.7	1.4	0.9	56.6	68.6	223.0	VE/AL	globulin 3; 18-kD globulin	protein
Zm00001d048806	0.1	0.1	1.1	10.4	38.0	61.5	VE/AL	22-kD zein	protein
Zm00001d020591	2.4	3.9	158.5	127.3	158.5	455.5	VE/SE	50-kD zein	protein

Table S9 in our manuscript. The representation of enriched genes involved in lipid, starch, and protein pathways in this study.

Figure 2f in our manuscript. The enriched genes from GO and WGNCA analysis involved in the pathways of lipid, starch and protein are highly expressed compared with other non-marker genes.

(5) The logic between different sections is somewhat confusing. For example, in the section "Mining key genes essential for sucrose transport and storage accumulation" the author proposed "These results together confirmed that the technology of spatial transcriptomics is more sensitive and robust than the experimental RNA in situ hybridization". I think this conclusion should be put forward when the author introduces the data quality of 10x Genomics Visium to emphasize the technical advantages of spatial transcriptome.

In response to the reviewer's comments, we have revised the logic and placement of the statement comparing spatial transcriptomics with experimental RNA in situ hybridization into Introduction.

REVIEWERS' COMMENTS

Reviewer #1 (Remarks to the Author):

The authors have addressed my concerns. I have no more comments.